# Hybrid CNN and XGBoost Model Tuned by Modified Arithmetic Optimization Algorithm for COVID-19 Early Diagnostics from X-ray Images

**Miodrag Zivkovic** [1,*] , **Nebojsa Bacanin** [1,*] , **Milos Antonijevic** [1] , **Bosko Nikolic** [2] , **Goran Kvascev** [2] , **Marina Marjanovic** [1] and **Nikola Savanovic** [1]

1 Faculty of Informatics and Computing, Singidunum University, 11010 Belgrade, Serbia
2 School of Electrical Engineering, University of Belgrade, 11120 Belgrade, Serbia
* Correspondence: mzivkovic@singidunum.ac.rs (M.Z.); nbacanin@singidunum.ac.rs (N.B.)

**Abstract:** Developing countries have had numerous obstacles in diagnosing the COVID-19 worldwide pandemic since its emergence. One of the most important ways to control the spread of this disease begins with early detection, which allows that isolation and treatment could perhaps be started. According to recent results, chest X-ray scans provide important information about the onset of the infection, and this information may be evaluated so that diagnosis and treatment can begin sooner. This is where artificial intelligence collides with skilled clinicians' diagnostic abilities. The suggested study's goal is to make a contribution to battling the worldwide epidemic by using a simple convolutional neural network (CNN) model to construct an automated image analysis framework for recognizing COVID-19 afflicted chest X-ray data. To improve classification accuracy, fully connected layers of simple CNN were replaced by the efficient extreme gradient boosting (XGBoost) classifier, which is used to categorize extracted features by the convolutional layers. Additionally, a hybrid version of the arithmetic optimization algorithm (AOA), which is also developed to facilitate proposed research, is used to tune XGBoost hyperparameters for COVID-19 chest X-ray images. Reported experimental data showed that this approach outperforms other state-of-the-art methods, including other cutting-edge metaheuristics algorithms, that were tested in the same framework. For validation purposes, a balanced X-ray images dataset with 12,000 observations, belonging to normal, COVID-19 and viral pneumonia classes, was used. The proposed method, where XGBoost was tuned by introduced hybrid AOA, showed superior performance, achieving a classification accuracy of approximately 99.39% and weighted average precision, recall and F1-score of 0.993889, 0.993887 and 0.993887, respectively.

**Keywords:** convolutional neural networks; COVID-19; metaheuristics; optimization; arithmetic optimization algorithm; sine cosine algorithm; XGBoost

## 1. Introduction

The COVID-19 pandemic has resulted in a huge worldwide catastrophe and has had a substantial impact on many lives across the world. The first instance of this deadly virus was reported in December 2019 from Wuhan, a Chinese province in [1]. After emergence, the virus quickly became a worldwide epidemic, impacting many nations across the globe. Reverse transcription-polymerase chain reaction (RT PCR) is one of the most often utilized methods in the diagnosis of COVID-19. However, since PCR has a diagnostic sensitivity of about 60–70%, radiological imaging techniques including computed tomography (CT) and X-ray have been critical in the early detection of this disease [2]. Therefore, the COVID-19 diagnosis from CT and X-ray images is an active and promising research domain, and additionally, there is much more space for improvements

A few recent investigations have found alterations in X-ray and CT imaging scans in individuals with COVID-19 symptoms. For example, Zhao et al. [3] discovered dilatation

and consolidation, as well as ground-glass opacities, in COVID-19 patients. The fast increase in the number of positive COVID-19 instances has heightened the necessity for researchers to use artificial intelligence (AI) alongside expert opinion to aid doctors in their work. Deep learning (DL) models have begun to gain traction in this respect. Due to a scarcity of radiologists in hospitals, AI-based diagnostic models may be useful in providing timely assistance to patients. Numerous research studies based on these approaches have been published in the literature; however, only the notable ones are mentioned here. Hemdan et al. [4] suggested seven convolutional neural network (CNN) models, including enhanced VGG19 and Google MobileNet, to diagnose COVID-19 from X-ray pictures. Wang et al. [5] classified COVID-19 pictures from normal and viral pneumonia patients with an accuracy of 92.4%. Similarly, Ioannis et al. [6] attained a class accuracy of 93.48% using 224 COVID-19 pictures. The Opconet, an optimized CNN, was proposed in [7] utilizing a total of 2700 pictures, giving an accuracy score of 92.8%. Apostolopoulous et al. [8] created a MobileNet CNN model utilizing extricated features. In [9], three different CNN models, namely inception v3, ResNet50, and Inception-ResNet V2, were employed for classification. In [10], a transfer learning-based method was utilized to classify COVID and non-COVID chest X-ray pictures utilizing three models such as ResNet18, ResNet50, SqueezeNet, and DenseNet121.

Although all of the above-mentioned state-of-the-art approaches use CNN, the methods do not take into consideration the spatial connections between picture pixels when training the models. As a result, when the pictures are rotated, certain resizing operations are performed, and data augmentation is executed owing to the availability of lower dataset sizes, the generated CNN models fail to properly distinguish COVID-19 instances, viral pneumonia, and normal chest X-ray scans. Although some degree of inaccuracy in recognizing viral pneumonia cases is acceptable, the misclassification of COVID-19 patients as normal or viral pneumonia might confuse doctors, leading to failure of early COVID-19 detection.

One of the promising ways for establishing an efficient COVID-19 detection model based on DL is to generate a network with proper architecture for each COVID-19 dataset. The no free lunch theorem (NFL) [11], which claims that the universal method for tackling all real-world problems does not exist, proved as right in the DL domain [12] and consequently standard DL model cannot render performance as good as models specifically tuned for COVID-19 diagnosis. The challenge of finding appropriate CNN and DL structures for each particular task is known in the literature as CNN (DL) hyperparameters tuning (optimization), and a good way to do it is by using an automated approach guided by metaheuristics optimizers [12–20]. The metaheuristics-driven CNN tuning has also been successfully applied to COVID-19 diagnostics [21–24].

However, the CNN tuning via metaheuristics is extremely time consuming because every function evaluation requires a generated network to be trained on large datasets for measuring solutions' quality (fitness). Additionally, the CNN training process with standard algorithms, e.g., gradient descent (GD) [25], conjugate gradienton (CG) [26], Krylov subspace descent (KSD) [27], etc., itself is very slow, and it can take hours to obtain feedback. Taking into account that the COVID-19 diagnostics is critical and that the efficient network needs to be established in almost real time, more approaches for COVID-19 early detection from X-ray and CT images are required.

With the goal of shortening training time, while performing automated feature extraction, research presented in this manuscript adapts a sequential, two-phase hybrid machine learning model for COVID-19 detection from X-ray images. In the first phase, a well-known simple architecture alike LeNet-5 CNN [28] is used as the feature extraction to reduce structural complexities within images. The second phase uses extreme gradient boosting (XGBoost) for performing classification, where outputs from the flatten layer of the LeNet-5 structure are used as XGBoost inputs. In other words, LeNet-5 fully connected (FC) layers are replaced with XGBoost to perform almost real-time classification. The LeNet structure

is trained only once, shortening execution time substantially more than in the case of CNN tuned approaches.

However, according to the NFL, the XGBoost, which efficiency depends on many hyperparameters, also needs to be tuned for specific problems. Consequently, this study also proposes metaheuristics to improve XGBoost performance for COVID-19 X-ray images classification. For the purpose of this study, modified arithmetic optimization algorithm (AOA) [29], that represents a low-level hybrid between AOA and sine cosine algorithm (SCA) [30], is developed and adapted for XGBoost optimization. The observed drawbacks of basic AOA are analyzed, and a method that outscores the original approach is developed. This particular metaheuristics is chosen because it shows great potential in solving varieties of real-world challenges [31,32]; however, since it relatively recenty emerged, it is still not investigated enough, and there are still many open spaces for its improvements .

The proposed two-phases hybrid method for COVID-19 X-ray diagnosis is validated against the COVID-19 radiography database set of images, which was retrieved from the Kaggle repository [33,34]. The classification is performed against three classes, namely normal, COVID-19 and viral pneumonia. The viral pneumonia X-rays are also taken because only subtle differences with COVID-19 X-ray images exist. However, since the source of the COVID-19 X-ray diagnosis dataset is imbalanced toward the normal class and the aim of the proposed research is not oriented toward addressing imbalanced datasets, the COVID-19 and viral pneumonia images are augmented, while the normal images are contracted from the original repository, and at the end each class, they contained 4000 observations.

The performance of the proposed methodology is compared with other standard DL methods as well as with XGBoost classifiers tuned with other well-known metaheuristics. Additionally, the proposed modified AOA, before being adopted for XGBoost tuning for COVID-19 classification, was first tested in optimizing challenging congress on evolutionary computation 2017 (CEC2017) benchmark instances.

Considering the above, this manuscript proposes a method that is guided by the two elemental problems for investigation:

- *The possibility of designing a method for efficient COVID-19 diagnostics from X-ray images based on the simple CNN and XGBoost classifier* and
- *The possibility of further improving the original AOA apporach by performing low-level hybridization with SCA metaheuritiscs.*

Established upon the experimental findings showed in Sections 4 and 5, the contribution of the proposed study is four-fold:

- A simple light-weight neural network has been generated that obtains a decent level of performance on the COVID-19 dataset and executes fast;
- An enhanced version of AOA metaheuristics has been developed that specifically targets the observed and known limitations and drawbacks of the basic AOA implementation;
- It was shown that the proposed metaheuristics is efficient in solving global optimization tasks with combined, real and integer parameters types; and
- The proposed COVID-19 detection methodology from X-ray images that employs the light-weight network, XGBoost and enhanced AOA obtains satisfying performance within a reasonable amount of computational time.

The sections of the manuscript are outlined as follows: Section 2 provides a brief survey of the AI method employed in this study with a focus on CNN applications. Section 3 explains the basic version of the AOA, points out its drawbacks and introduces the modified AOA implementation. Bound constrained simulations of the proposed algorithm on a challenging CEC2017 benchmark set are given in Section 4. The experimental findings of the COVID-19 early diagnostics from X-ray images with the proposed methodology are provided in Section 5, while the final remarks, proposed future work and conclusions are given in Section 6.

## 2. Background and Preliminaries

The following section aims to give a theoretical background for the used methods and to elaborate the workings of the proposed method, which is described later on. Firstly, the deep neural networks (DNN) will be explained alongside an emphasis on CNN. Afterwards, the XGBoost architecture is described followed by the metaheuristic optimization.

### 2.1. Deep Neural Networks

The application of deep learning models to the analysis of X-ray captures is widely applied [35–37]. The performance of CNN is distinguished among deep learning models, and that is the case for X-ray image classification as well [38,39]. Input is transformed through many layers of the CNN and the application of narrow filters. The types of CNNs are various, and some of interest for this research are: ResNet [40], AlexNet [41], ZFNet [42], VGGNet [43], GoogLeNet [44], and LeNet-5 [45]. Considering the prolonged evolutionary process of the metaheuristics optimizer that is suggested for this part of the solution, the contribution of large networks can be considerable in terms of computational costs [46]. Nevertheless, operating with large networks can result in overfitting [47].

Black and white as well as grayscale images are best used with the models such as LeNet, with advantages that include simplicity alongside effectiveness. To increase the real-time processing capabilities, the authors propose the use of simple network structure such as LeNet as the primary classifier, for the structural complexity reduction and out of consideration to the previously mentioned limitations. Introduced by Yann Le-Cun, the LeNet-5 [28] is considered the simplest from the family of CNNs, and its architecture is presented in the Figure 1. This network includes only two convolutional and average pooling layers, while it uses three fully connected layers for output classification/regression.

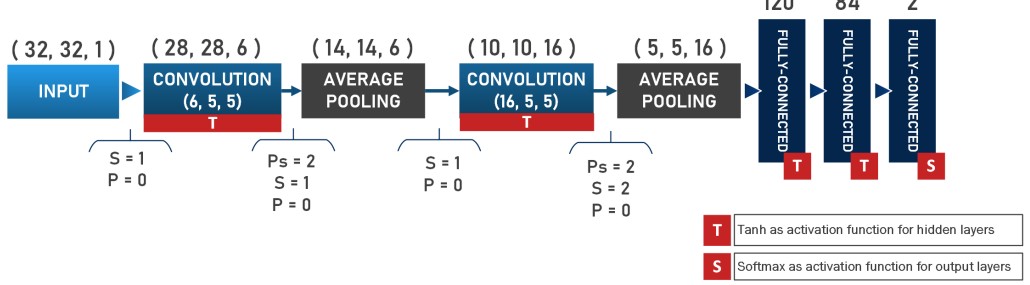

**Figure 1.** The design of LeNet-5 CNN.

The visual tasks heavily employ the CNN technology [48], with contemporary progress in the field of facial recognition [49,50], analysis of documents [51,52], classification of medical images and diagnostics [53–55], as well as a paramount task of climate change analysis and severe weather conditions [56,57] including various other applications. In spite of the diversity of its practical implementation, the CNNs are not perfect. A considerable task is the overfitting issue and methods to avoid it. The popular solutions are regularization and dropout, early stopping, model simplification, and data augmentation [58–62]. The approach that the authors focused on is the dropout [63,64]. The principle of removing a unit from a layer including its connections is referred to as dropping. The selection of units to be dropped is random, and they are temporarily removed during the process of training. The neurons are detached so the network would achieve better generalization, and it does so as a result of desensitization to the neurons weights. To achieve an optimal weight set in a polynomial time is an NP-hard problem [15,65].

The architecture of the CNNs is a layer based on the goal of mimicking the human visual cortex. The types of these layers are convolutional, pooling and dense. The input goes through all layers in a specific order, which results in a high level of features allowing for high-precision image classification and optimization. The loss function has to be optimized during the weight learning of network training, and some of the optimizers are adadelta,

adagrad, adamax, Adam, rmsprop, stochastic gradient descent, and momentum [66–68]. Non-linear output is mapped through the transfer (activation) function, and examples of such functions are rectified linear unit (ReLU) [69], tanh, and sigmoid. The de facto standard has been achieved by the ReLu transfer unction with a value amounting to $f(x) = max(x, 0)$.

Hyperparameters heavily influence the accuracy of the model and are a key subject of optimization [13]. The number of kernels and kernel size of each convolutional layer, the learning rate, the batch size, the number of convolutional and fully connected (dense) layers, the weight regularization in the dense layers, the activation function, the dropout rate, and so on are some examples of hyperparameters. Hyperparameter optimization is not a process that can be universally solved across all problems; hence, the "trial and error" approach is necessary. Such methods can be time exhaustive and do not guarantee results. This process is deemed as NP-hard. Metaheuristic methods have yielded results with such endeavors [70–72].

The detailed CNN mathematical formulation is provided in [73], and a more recent study on the same topic is given in [74].

### 2.2. The XGBoost Algorithm

An adaptive training method is used by the XGBoost algorithm for objective function optimization. Consequentially, every step in the optimization process depends on the previous step in terms of the result. The mathematical expression of the objective function of the XGBoost model is listed below:

$$F_o{}^i = \sum_{k=1}^n l\left(y_k, \hat{y}_k^{i-1} + f_i(x_k)\right) + R(f_i) + C, \tag{1}$$

where the $t$-th iteration loss term is given as $l$, the constant term is $C$, and the regularization parameter $R$ of the model is additionally described as:

$$R(f_i) = \gamma T_i + \frac{\lambda}{2} \sum_{j=1}^T w_j^2 \tag{2}$$

Generally, the simplicitty of the tree structure is proportional to the value of $\gamma$ and $\lambda$ customization parameters. The larger the value of the parameters is, the simpler the tree structure. First and second derivatives of the model, $g$ and $h$, respectively, are given as follows:

$$g_j = \partial_{\hat{y}_k^{i-1}} l\left(y_j, \hat{y}_k^{i-1}\right) \tag{3}$$

$$h_j = \partial^2_{\hat{y}_k^{i-1}} l\left(y_j, \hat{y}_k^{i-1}\right) \tag{4}$$

The following formulas are used for obtaining the solution:

$$w_j^* = -\frac{\sum g_t}{\sum h_t + \lambda} \tag{5}$$

$$F_o{}^* = -\frac{1}{2} \sum_{j=1}^T \frac{(\sum g)^2}{\sum h + \lambda} + \gamma T, \tag{6}$$

where the loss function score is given as $F_o{}^*$, while the solution weights are given as $w_j^*$.

### 2.3. Metaheuristic Optimization

Metaheuristics optimization algorithms are stochastic approaches that can be utilized to solve NP-hard problems where deterministic algorithms cannot obtain the solution in a reasonable amount of time with a reasonable amount of resources. Several families of algorithms exist in this group; however, different authors classify them in a different way. One of the most commonly used taxonomies divides metaheuristics with respect to the type of phenomena that was used to model the search mechanism of the algorithm [75–77]. This categorization divides metaheuristics approaches into swarm intelligence and genetic

algorithms (both being inspired by the nature and a variety of behavior exhibited by animals in large groups), algorithms inspired by physical processes (such as gravitational search, water waves or electromagnetism), human-based algorithms (social network behavior, teaching, learning, and brainstorming process for example) and the most recent group of algorithms inspired by the mathematical properties (sine, cosine, arithmetic operations and so on).

The field of swarm intelligence belongs to the group of metaheuristic algorithms that apply the behavior of animals that live in swarms to the algorithms that are used in the domain of artificial intelligence [78,79]. This type of algorithm has proven efficient in tackling NP-hard problems for a large variety of applications. The true potential of swarm intelligence algorithms is accomplished through the process of hybridization. With the use of this method, the convergence speed can be substantially increased. The foundation is the stochastic methodology with the search mechanism for global optima. This results in heavy reliance on the amount of iterations. The search process recognizes two different phases alike with the training and testing phases in machine learning. These two phases are exploration, which is focused on searching locally, and exploitation, which is directed toward global search. The problem is balancing these two phases. Swarm algorithms are not expected to provide the certainly best solution but rather a very close one to it referred to as sub-optimal. Evolutionary principles immensely improve the search process if applied to the algorithm. The idea is to transfer the information from the current population to the following one. Evolution recognizes three different operations: selection, mutation, and crossover. The simplest one is the selection of the best units and using them in their original form in the next population. The same is completed with the mutation process but with some changes to the value that the unit carries over. Finally, the crossover operation combines two units. The most acclaimed SI solutions consist of ant colony optimization (ACO) [80], bat algorithm (BA) [81,82], (PSO) [83], artificial bee colony (ABC) [84], firefly algorithm (FA) [85], and a more recent quantum-based avian navigation optimizer [86].

Even though the mentioned algorithms have individual high performance, the hybrid solutions still outperform them. The trend of hybridization is increasing, and the researchers gravitate toward modified solutions. Noteworthy examples of these algorithms are the ABC-BA [87], interactive search algorithm (ISA) [88], Swarm-TWSVM [89], and two-stage GA-PSO-ACO algorithm [90].

The most recent group of metaheuristics algorithms draws inspiration from the mathematical processes and laws. Two of the most significant representatives of this group are the sine–cosine algorithm (SCA) [30] and arithmetic optimization algoritm (AOA) [29]. The SCA algorithm is inspired by mathematical fluctuations of the sine and cosine functions, while the AOA utilizes fundamental mathematical operators, and both of them were utilized in the approach suggested in this paper. There are also other recently proposed algorithms that fall into this group, including golden sine algorithm (Gold-SA) [91], for example.

The main obstacle with the use of population-based algorithms is natural to the machine learning field to which they belong, and that is that there is no universally best solution for all problems. The no free lunch (NFL) theorem provides the theory to support this claim [11]. Hence, the high diversity in algorithms and their versions so that every use case has the best adapted solution.

The real-life application of the population-based metaheuristics algorithms is various, and some of them are cloud computing [92–94], cloud-edge computing [95], wireless sensor networks [96–99], COVID-19 case number prediction [100,101], feature selection problem [102,103], classification of glioma MRI images [17], global optimization problems and engineering optimization [104–106], credit card frauds detection [107,108], pollution prediction [109] as well as general machine learning optimization [110,111].

The tuning of deep neural networks is an additional trend that has emerged in the field of swarm intelligence. As already mentioned, these algorithms have proven excellent performance in solving NP-hard problems. This problem with the DNN emerges with

hyperparameter optimization, and the swarm algorithms have solved this problem in countless different cases [4–6].

The XGBoost method used in this work also had its fair share of improvements through metaheuristic optimization. Notable cases of these types of solutions are: [112], which tests the classification of different metaheuristic approaches alongside XGBoost, ref. [113] applies PSO to the problem of network intrusion, and [114] for stock price prediction utilizing XGBoost and genetic algorithm (GA). Additionally, XGBoost tuned by the metaheuristics was used in intrusion detection and network security models [115–118].

## 3. Proposed Methodology

This section first shows a brief overview of original AOA metaheuristics, which is followed by its observed drawbacks and devised modified hybrid metaheuristics approach for the purpose of this study. Finally, this section concludes with a presentation of the two-phase sequential DL and XGboost method, which is used for COVID-19 X-ray images categorization.

### 3.1. Arithmetic Optimization Algorithm

A novel method called arithmetic optimization algorithm (AOA) is a metaheuristic method which draw inspiration from mathematics fundamental operators introduced by Abuligah et al. [29].

The optimization process of AOA initializes with $X$, a randomly generated matrix, for which the single solution is represented as $X_{ij}$, $1 \leq i \leq N$, and $1 \leq j \leq n$, which represents the initial optimization space for solutions. The best-obtained solution is decided after each iteration and is considered a candidate for the best solution. The operations subtraction, addition, division, and multiplication control the computation of the near-optimal solution areas. The search phase selection is calculated according to the Math Optimizer Accelerated (MOA) function applied during both phases:

$$MOA(t) = Min + t \times \left( \frac{Max - Min}{T} \right) \tag{7}$$

where the $t$-th iteration function value is given as $MOA(t)$, while the range is 1 to the maximum iterations number $T$ in which the current iteration is signified as $t$. $Min$ and $Max$, respectively, represent the minimum and maximum accelerated function values.

The search space is randomly explored with the use of division ($D$) and multiplication ($M$) operators during the exploration phase. This mechanism is given with Equation (8). When the condition $r1 > MOA$ is satisfied, the search is limited by the MOA for the current phase. The operator ($M$) will not be applied until the first operator ($D$) does not finish its task conditioned by $r2 < 0.5$ as the first rule of Equation (8). Otherwise, operator $D$ is substituted by the ($M$) operator for the completion of the same task.

$$X_{i,j}(t+1) = \begin{cases} best(X_j) \div (MOP + \epsilon) \times \left( (UB_j - LB_j) \times \mu + LB_j \right), & r2 < 0.5 \\ best(X_j) \times MOP \times \left( (UB_j - LB_j) \times \mu + LB_j \right), & \text{otherwise} \end{cases} \tag{8}$$

where the arbitrary small integer is $\epsilon$, the fixed control parameter is $\mu$, the $i$-th solution of the next iteration is $X_{i,j}(t+1)$, the current location $j$ of the current iteration's $i$-th solution is $X_{i,j}(t)$, and the current best solution's $j$-th position is $best(X_j)$. Standardly, the lower and upper boundaries of the $j$-th position are $LB_j$ and $UB_j$.

$$MOP(t) = 1 - \frac{t^{1/\alpha}}{T^{1/\alpha}} \tag{9}$$

where the $t$-th iteration function value is denoted as the Math Optimizer Probability $MOP(t)$, the current iteration is $t$, the maximum iterations number is $T$, and the fixed parameter is $\alpha$ with the purpose of measuring the accuracy of exploitation over iterations.

The deep search of the search space for exploitation is afterwards performed by the search strategies employed with addition (*A*) and subtraction (*S*) operators. This process is provided in Equation (10). The bounds of the first rule of Equation (10) are $r3 < 0.5$ which similarly links the operator (*A*) to the operator (*S*) as in the previous phase as (*M*) to (*D*). Furthermore, (*S*) is substituted by (*A*) to finish the task,

$$X_{i,j}(t+1) = \begin{cases} best(X_j) - MOP \times \left( (UB_j - LB_j) \times \mu + LB_j \right), & r3 < 0.5 \\ best(X_j) + MOP \times \left( (UB_j - LB_j) \times \mu + LB_j \right), & \text{otherwise} \end{cases} \qquad (10)$$

Conclusively, the near-optimal solution candidates tend to diverge when $r1 > MOA$, while they gravitate to near-optimal solutions in case of $r1 < MOA$. For the stimulation of exploration and exploitation, the values from 0.2 to 0.9 are incrementally increased for the $MOA$ parameter. Additionally, note that the computational complexity of AOA is $O(N \times (ML + 1))$ computational complexity.

### 3.2. Cons of Basic AOA and Introduced Modified Algorithm

The basic version of the AOA is regarded as a potent optimizer with a wide range of practical applications, but it stills suffer from several known drawbacks in its original implementation. These flaws are namely insufficient exploitation power and an inadequate intensity of exploration process. This is reflected in the fact that in some cases, AOA is susceptible to dwell in the proximity of the local optima and also to the slow converging speed [32,119,120], as it can clearly be observed in CEC2017 simulations presented in Section 4.

One of the root causes of these deficiencies is that the solutions' update procedure in basic AOA is focused on the proximity of the single current global best solution. As discussed by [119,121], it results in an extremely selective search procedure, where other solutions depend on the solitary centralized guidance to update their position, with no guarantees to converge to the global optimum. Hence, it is necessary to improve the exploration capability of the basic AOA to escape the local optimums.

Due to the above-mentioned cons, during the search process, the original AOA converges too fast toward the current best solution, and the population diversity is disturbed. Since the AOA's efficiently depends to some extent on the generated pseudo-random numbers due to its stochastic nature, in some runs, when the current best individual in the initial population is close to optimum regions of the search domain, the AOA shows satisfying performance. However, when the algorithm is "unlucky" and the initial population is further away from optimum, the whole population quickly converges toward sub-optimum regions, and the final results have lower quality.

Additionally, besides poor exploration, the AOA's intensification process can be also improved. As already noted, the search is conducted mostly in the neighborhood of the current best individual, and exploitation around other solutions from the population is not emphasized enough.

The enhanced AOA proposed in this manuscript addresses both observed drawbacks by improving exploration, exploitation and its balance of the original version. For that reason, the proposed method introduces the search procedure from another metaheuristics and an additional control parameter that enhances exploration, but it also establishes better intensification–diversification trade-off.

The authors were inspired by the low-level methodology of hybridization employing the principles from SCA to the AOA. This process results in satisfactory performance from both phases of the metheuristic solutions and a superior hybrid solution. The basic equations for position updating with the SCA are given (11):

$$X_i^{t+1} = \begin{cases} X_i^t + r_1 \times \sin(r_2) \times |r_3 P_i^t - X_i^t| & r_4 < 0.5 \\ X_i^t + r_1 \times \cos(r_2) \times |r_3 P_i^t - X_{i_i}^t| & r_4 \geq 0.5 \end{cases} \qquad (11)$$

where the current option's setting for the *i*-th measurement at the *t*-th model is $X_i^t$, arbitrary numbers $r_1/r_2/r_3$, the location factor placement in the *i*-th dimension is $P_i$, and the absolute value is given as $||$.

As stated above, after conducting extensive examination of the search equations of AOA and SCA algorithms, it was determined that AOA search equations are not sufficient for efficient exploitation, which to a large extent depends on the current best solution, and it is required to cover a wider search space. Hence, this research aimed to merge two algorithms combined with using a quasi-reflection-learning based (QRL) procedure [122] in the following way. Every solution life-cycle consists of two phases, where the solution performs an AOA search (phase one) and SCA search (phase two), which are controlled by the value of one additional control parameter.

Each solution is assigned a *trial* attribute, which is utilized to monitor the improvement of the solutions. In the beginning, after producing the initial population, all solutions start with an AOA search. In each iteration, if the solution was not improved, the *trial* parameter is increased by 1. When *trial* reaches the threshold value *limit* (control parameter in the proposed hybrid algorithm), that particular solution continues the search by switching to the SCA search mechanism. Again, every time when the solution is not improved, *trial* is increased by 1. If the *trial* reaches the $2 \cdot limit$ value, that solution is removed from the population and replaced by the quasi-reflexive-opposite solution $X^{qr}$ of the solution $X$, which is generated by applying Equation (12) over each component *j* of solution *X*.

$$X^{qr} = \mathrm{rnd}\left(\frac{LB + UB}{2}, X\right),$$ (12)

where $\mathrm{rnd}\left(\dfrac{LB + UB}{2}, X\right)$ part of the equation has a role to generate a random value derived from the uniform distribution inside $\left[\dfrac{LB + UB}{2}, X\right]$, and *LB* and *UB* represent the lower and upper limits of the search space, respectively. This procedure is executed for each parameter of every solution *X* within *D* dimensions.

However, the replacement is not performed for the current best solution, because, practically, if the solution manages to maintain the best rank within $2 \cdot limit$ iterations, there is a great chance that this solution hits the right part of the search space. If such a replacement would have occured, then the search process might diverge from the optimum region.

It must be noted that when replacing the solution with its opposite, additional evaluation is not performed. The logic behind utilizing the quasi-reflexive opposite solutions is based on the fact that if the original solution did not improve for a long time, it was located far away from the optimum (or in one of the sub-optimum domains), and there is a reasonable chance that the opposite solution will fall significantly closer to the optimum. Discarding so-called exhausted solutions from the population ensures stable exploration during the whole search process in the run. The novel solution starts its life-cycle as described above, with the *trial* parameter reset to 0, and by conducting the AOA search first.

The value of the *trial* threshold was determined empirically, and it is calculated by using the following expression: $limit = \frac{T}{2 \cdot N}$, where *T* denotes the maximal number of iterations, and *N* is the size of the population. Therefore, there is no need for the researcher to fine-tune this parameter.

For simplicity reasons, the introduced AOA method is named hybrid AOA (HAOA) and its pseudo-code is provided in Algorithm 1. The introduced changes do not increase the complexity of the original AOA algorithm; hence, the complexity of the proposed HAOA is estimated as $O(N) = N + N \cdot T$. Moreover, the HAOA introduces just one additional control parameter (*limit*), and it is automatically determined as it depends on *T* and *N*.

---

**Algorithm 1:** Hybrid arithmetic optimization algorithm.

---

Initialize the parameters $\alpha$ and $\mu$.
Initialize solutions' positions randomly ($i = 1, ..., N$).
Set *trial* values of each solution to 0.
Determine *limit* value as $limit = \frac{T}{2N}$
**while** $t < T$ **do**
   Compute the fitness function for the given solutions.
   Find the best solution so far.
   Update MOA and MOP values using Equations (7) and (9), respectively.
   **for** $i = 1$ to *Solutions* **do**
     **if** *trial* < *limit* **then**
       Execute AOA search
       **for** $j = 1$ to $D$ **do**
         Generate a random number ($r1, r2, r3$) in interval [0, 1].
         **if** $r1 > MOA$ **then**
           **Exploration phase**
           **if** $r2 > 0.5$ **then**
             Apply the division operator ($D$, "÷")
             Update the $i$th solutions' positions using the first rule in Equation (8).
           **else**
             Apply the multiplication operator ($M$, "×")
             Update the $i$th solutions' positions using the second rule in Equation (8).
           **end if**
         **else**
           **Exploitation phase**
           **if** $r3 > 0.5$ **then**
             Apply the subtraction operator ($S$, "−")
             Update the $i$th solutions' positions using the first rule in Equation (10).
           **else**
             Apply the addition operator ($A$, "+")
             Update the $i$th solutions' positions using the second rule in Equation (10).
           **end if**
         **end if**
       **end for**
       Compare the old solution and updated solution and increment *trial* if needed.
     **else if** *trial* < 2 ∗ *limit* **then**
       Execute SCA search
       **for** $j = 1$ to $D$ **do**
         Update positions according to Equation (11).
       **end for**
       Compare old solution and updated solution and increment *trial* if needed.
     **else**
       **if** $i$ is not the current best solution **then**
         Remove solution $X_i$ from the population.
         Replace $X_i$ with quasi-reflexive-opposite solution $X_i^{qr}$ produced with Equation (12).
         Reset *trial* parameter to value 0.
       **end if**
     **end if**
   **end for**
   $t = t + 1$
**end while**
Return the best solution.

---

### 3.3. Deep Learning Approach for Image Classification

As is it was already mentioned in Section 1, the proposed approach is executed in two phases, where the first phase performs feature extraction and the second phases employs XGBoost for performing classification.

In the first phase of the proposed approach, a simple CNN architecture, similar to LeNet5 [28] that consists of 3 convolutional and 3 max pooling layers, followed by 3 fully-connected layers, is employed. This network structure was determined empirically with the goal of being as simple as possible (allowing easier training and fast execution), while achieving a decent level of performance on the COVID-19 dataset, by performing hyperparameters optimization during the pre-research phase via a simple grid search. The hyperparameters that were tuned included the number of convolutional layers (range $[2, 5]$, integer), number of cells in convolutional layers (range $[3, 36]$, integer), number of fully connected layers (range $[2, 5]$, integer) and learning rate (range $[0.00001, 0.1]$, continuous). The determined network structure is as follows: the first convolutional layer uses 32 filters with $3 \times 3$ kernel size, while the second and third convolutional layers employ 16 filters with $3 \times 3$ kernels, which is followed by 3 dense layers. The complete CNN network structure is shown in Figure 2.

All images are resized to $32 \times 32$ pixel size and used as CNN input, where the input size is $32 \times 32 \times 3$. The convolutional layers' weights are pre-trained on a COVID-19 dataset, as described in Section 5.1 with the Adam optimizer and a learning rate ($\eta$) of 0.001, *sparseCatagoricalCrossEntropy* loss function and a batch size of 32 over 100 epochs. The CNN uses a training set and validation set, which is a 10% fraction of the training data, and an early stopping condition with respect to validation loss with patience set to 10 epochs.

Due to the stochastic nature of the Adam optimizer, the whole training process is repeated 50 times, and the best performing pre-training model is used for the second phase. Training and validation loss for the best model during the training is shown in Figure 3, where it can be seen that the due to early stopping criteria, training terminated after only 60 epochs.

After determining the sub-optimal weights and biases of the used simple CNN in the first phase, in the second phase, all fully connected layers from the CNN are removed, and the outputs from CNN's flatten layer are used as inputs for the XGBoost classifier. Therefore, all CNN's fully connected layers are replaced with XGBoost, where XGBoost inputs represent features extracted by the convolutional and maxpooling layers of CNN.

However, as it was also pointed out in Section 1, the XGBoost should be optimized for every particular dataset. Therefore, the proposed HAOA is used for XGBoost tuning, where each HAOA solution is of length 6 ($L = 6$), with every solution's component representing one the XGBoost hyperparameters.

The collection of XGBoost hyperparameters that were addressed and tuned in this research is provided below, together with their boundaries and variable types:

- Learning rate ($\eta$), limits: $[0.1, 0.9]$, category: continuous;
- *Min_child_weight*, limits: $[0, 10]$, category: continuous;
- Subsample, limits: $[0.01, 1]$, category: continuous;
- Collsample_bytree, limits: $[0.01, 1]$, category: continuous;
- Max_depth, limits: $[3, 10]$, category: integer; and
- *Gamma*, limits: $[0, 0.5]$, category: continuous.

The parameter count required by *softprob objective function* ('num_class':self.no_classes) is further being passed as the parameter to XGBoost as well. All other parameters are determined and set to default XGBoost values.

Finally, the hybrid proposed approach is named after the used models—CNN-XGBoost-HAOA, and its flowchart is depicted in Figure 4.

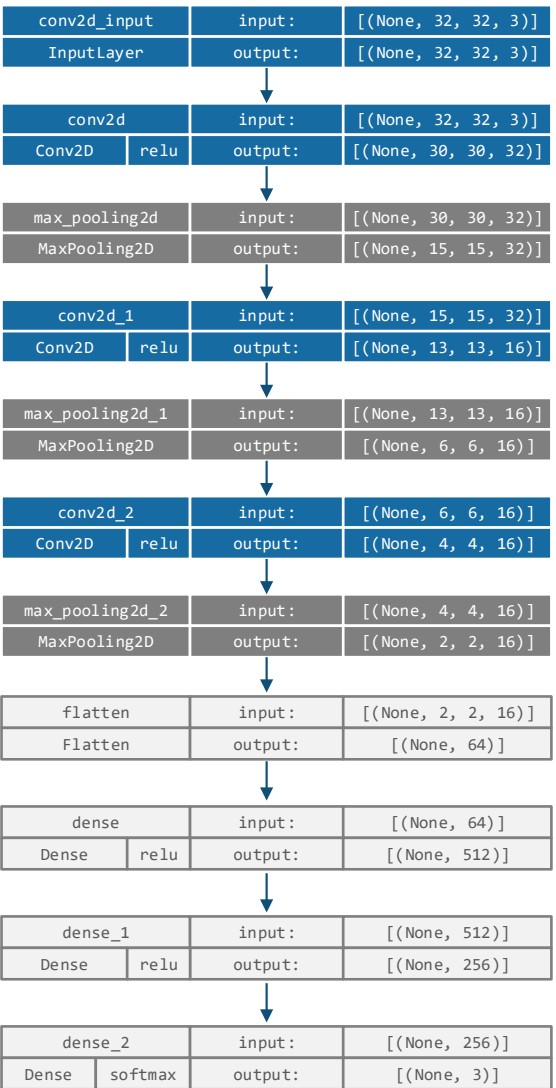

**Figure 2.** CNN structure used in the proposed approach.

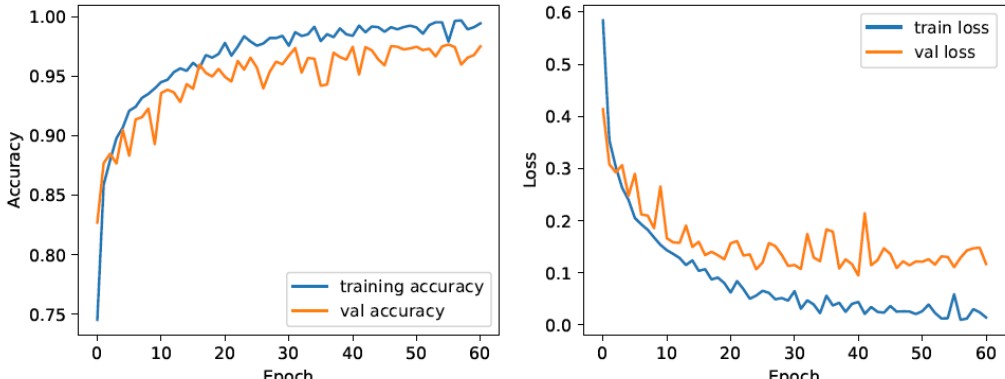

**Figure 3.** The CNN training model.

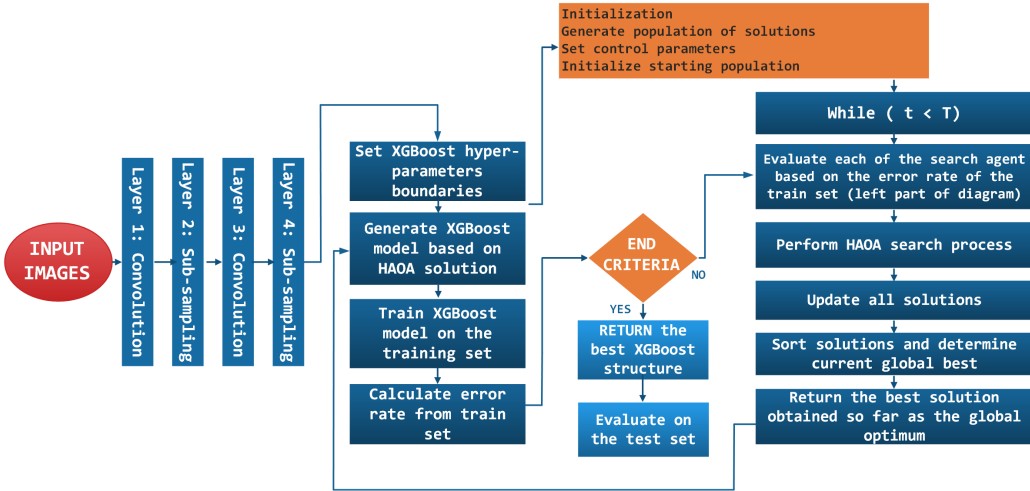

**Figure 4.** The CNN-XGBoost-HAOA flowchart.

## 4. CEC2017 Bound-Constrained Experiments

The XGBoost tuning belongs to the group of NP-hard global optimization problems with mixed, real values and integer parameters (see Section 3.3). However, to prove the robustness of the optimizer, it should be first tested on a larger set of global optimization benchmark instances before being validated against the practical problem such as XGBoost hyperparameters optimization.

Therefore, the HAOA was validated on exceedingly challenging global optimization benchmark functions from the CEC2017 testing suite [123] with 30 parameters. The total number of instances is 30, and they are divided into 4 groups: from *F*1 to *F*3—uni-modal, from *F*4 to *F*10—multi-modal, hybrid functions are instances from *F*11 to *F*20, and finally, the most challenging functions are the composite ones that include instances from *F*21 to *F*30. The composite benchmarks exhibit all characteristics of the previous 3 groups; plus, they have been rotated and shifted.

The *F*2 instance was discarded from experimentation due to its unstable behavior, as pointed out in [124]. The full specification of benchmark functions including name, class, parameters search range and global optimum value are shown in Table 1. More details, such as its visual representation, can be seen in [123].

All simulations were performed with 30-dimensional CEC2017 instances (*Dim* = 30), and results for obtained mean (average) and standard deviation (std) averaged over 50 separate runs are reported. These two metrics are the most representative due to the stochastic behavior of metaheuristics. A relatively extensive evaluation of metaheuristics performance for the CEC2017 benchmark suite is provided in [125], where state-of-the-art improved harris hawks optimization (IHHO) was introduced; therefore, a similar experimental setup as in [125] was used in this study.

The research proposed in [125] validated all approaches in simulations with 30 individuals in the population (*N* = 30) and 500 iterations (*T* = 500) throughout one runtime. However, some metaheuristics spare more *FFEs* in one run, and setting the termination condition in terms of iterations may not be the most objective strategy. Therefore, to compare the proposed HAOA with other methods without biases, and at the same time to be consistent with the above-mentioned study, this research uses 15,030 *FFEs* (*N* + *N* · *T*) as the termination condition.

Additionally, most of the methods presented for validation purposes in [125] were also implemented in this study with the same adjustments of control parameters. The comparison between the proposed HAOA and the following methods was performed: basic AOA, SCA, cutting-edge IHHO [125], HHO [126], differential evolution (DE) [127], grasshopper optimization algorithm (GOA) [128], gray wolf optimization (GWO) [129], moth flame

optimization (MFO) [130], multi-verse optimizer (MVO) [131], particle swarm optimization (PSO) [83] and whale optimization algorithm (WOA) [132].

**Table 1.** The CEC2017 benchamrk instances specifications.

| ID | Function's Name | Class | Search Range | Optimum |
|----|----------------|-------|--------------|---------|
| *F*1 | Shifted and Rotated Bent Cigar Function | Unimodal | [−100, 100] | 100 |
| *F*2 | Shifted and Rotated Sum of Different Power Function | Unimodal | [−100, 100] | 200 |
| *F*3 | Shifted and Rotated Zakharov Function | Unimodal | [−100, 100] | 300 |
| *F*4 | Shifted and Rotated Rosenbrock's Function | Multimodal | [−100, 100] | 400 |
| *F*5 | Shifted and Rotated Rastrigin's Function | Multimodal | [−100, 100] | 500 |
| *F*6 | Shifted and Rotated Expanded Scaffer's Function | Multimodal | [−100, 100] | 600 |
| *F*7 | Shifted and Rotated Lunacek Bi-Rastrigin Function | Multimodal | [−100, 100] | 700 |
| *F*8 | Shifted and Rotated Non-Continuous Rastrigin's Function | Multimodal | [−100, 100] | 800 |
| *F*9 | Shifted and Rotated Lévy Function | Multimodal | [−100, 100] | 900 |
| *F*10 | Shifted and Rotated Schwefel's Function | Multimodal | [−100, 100] | 1000 |
| *F*11 | Hybrid Function 1 ($N = 3$) | Hybrid | [−100, 100] | 1100 |
| *F*12 | Hybrid Function 2 ($N = 3$) | Hybrid | [−100, 100] | 1200 |
| *F*13 | Hybrid Function 3 ($N = 3$) | Hybrid | [−100, 100] | 1300 |
| *F*14 | Hybrid Function 4 ($N = 4$) | Hybrid | [−100, 100] | 1400 |
| *F*15 | Hybrid Function 5 ($N = 4$) | Hybrid | [−100, 100] | 1500 |
| *F*16 | Hybrid Function 6 ($N = 4$) | Hybrid | [−100, 100] | 1600 |
| *F*17 | Hybrid Function 6 ($N = 5$) | Hybrid | [−100, 100] | 1700 |
| *F*18 | Hybrid Function 6 ($N = 5$) | Hybrid | [−100, 100] | 1800 |
| *F*19 | Hybrid Function 6 ($N = 5$) | Hybrid | [−100, 100] | 1900 |
| *F*20 | Hybrid Function 6 ($N = 6$) | Hybrid | [−100, 100] | 2000 |
| *F*21 | Composition Function 1 ($N = 3$) | Composition | [−100, 100] | 2100 |
| *F*22 | Composition Function 2 ($N = 3$) | Composition | [−100, 100] | 2200 |
| *F*23 | Composition Function 3 ($N = 4$) | Composition | [−100, 100] | 2300 |
| *F*24 | Composition Function 4 ($N = 4$) | Composition | [−100, 100] | 2400 |
| *F*25 | Composition Function 5 ($N = 5$) | Composition | [−100, 100] | 2500 |
| *F*26 | Composition Function 6 ($N = 5$) | Composition | [−100, 100] | 2600 |
| *F*27 | Composition Function 7 ($N = 6$) | Composition | [−100, 100] | 2700 |
| *F*28 | Composition Function 8 ($N = 6$) | Composition | [−100, 100] | 2800 |
| *F*29 | Composition Function 9 ($N = 3$) | Composition | [−100, 100] | 2900 |
| *F*30 | Composition Function 10 ($N = 3$) | Composition | [−100, 100] | 3000 |

Results for the CEC2017 simulations are displayed in Table 2. The text in bold emphasizes the best results for every performance indicator and instance. In the case of equal performance, these results are also bolded. Regardless whether the experimentation in [133] was performed with *T* as the termination condition, the results reported in this study are similar. However, due to the stohastic behavior of the optimizer, subtle differences exist.

The best mean results for 21 functions were achieved by the HAOA, and they include *F*1, *F*3, *F*5, *F*6, *F*7, *F*8, *F*11, *F*12, *F*13, *F*15, *F*17, *F*19, *F*20, *F*21, *F*22, *F*23, *F*25, *F*26, *F*28, *F*29, and *F*30. The functions are shown in Table 2. The second best approach proved the best cutting-edge IHHO, and in some tests, the IHHO showed better performance than HAOA, while in others, the results of HAOA and IHHO were tied. The HAOA and IHHO obtained the same mean indicator values in the following tests: *F*3, *F*6, *F*19, *F*21, and *F*29. The small number of cases in which the HAOA performed worse than the IHHO includes *F*4 and *F*14 experiments. There are also some cases where other methods achieved the best results, e.g., the *F*9 instance, where MVO and PSO showed superior performance. Lastly, the HAOA tied DE in the cases of *F*13 and *F*15 instances.

Additionally, it is very important to observe that the original AOA never beat HAOA. Moreover, there are instances where the HAOA tremendously outscored AOA, even by more than 1000 times, e.g., in the function *F*1 test. Finally, it is also significant to compare HAOA and SCA, because the HAOA uses SCA search expressions. In all simulations, the HAOA outperformed SCA for both indicators. Accordingly, it can be concluded that the

HAOA successfully managed to combine the advantages of basic AOA and SCA methods as a low-level hybrid approach.

The magnitude of results' variances between the HAOA and every other method implemented in CEC2017 simulations can be determined from a Friedman test [134,135] and two-way ranks variance analysis. This was performed for the reasons of statistical importance of an improvement's proof that is more thorough than simply putting outcomes into comparison. Table 3 summarizes the results of the Friedman test over 29 CEC2017 instances for 12 compared methods.

**Table 2.** The CEC2017 results and comparative analysis—HAOA vs. others.

| Algorithm | F1 | | F2 | | F3 | | F4 | | F5 | |
|---|---|---|---|---|---|---|---|---|---|---|
| | Mean | STD | Mean | STD | Mean | STD | Mean | STD | Mean | STD |
| IHHO | $1.86 \cdot 10^2$ | 26.921 | n/a | n/a | $\mathbf{3.15 \cdot 10^2}$ | 52.152 | $\mathbf{4.03 \cdot 10^2}$ | 2.607 | $5.05 \cdot 10^2$ | **3.251** |
| HHO | $1.75 \cdot 10^6$ | $4.29 \cdot 10^5$ | n/a | n/a | $6.71 \cdot 10^2$ | $3.24 \cdot 10^2$ | $4.37 \cdot 10^2$ | 53.631 | $5.35 \cdot 10^2$ | 24.927 |
| DE | $7.54 \cdot 10^7$ | $1.71 \cdot 10^7$ | n/a | n/a | $4.59 \cdot 10^3$ | $1.35 \cdot 10^3$ | $4.29 \cdot 10^2$ | 8.530 | $5.52 \cdot 10^2$ | 6.232 |
| GOA | $1.56 \cdot 10^5$ | $5.24 \cdot 10^4$ | n/a | n/a | $3.18 \cdot 10^2$ | 61.300 | $4.15 \cdot 10^2$ | 19.48 | $5.25 \cdot 10^2$ | 16.803 |
| GWO | $1.53 \cdot 10^7$ | $4.85 \cdot 10^6$ | n/a | n/a | $3.57 \cdot 10^3$ | $2.77 \cdot 10^3$ | $4.09 \cdot 10^2$ | 10.705 | $5.19 \cdot 10^2$ | 8.543 |
| MFO | $7.17 \cdot 10^6$ | $2.18 \cdot 10^7$ | n/a | n/a | $9.04 \cdot 10^3$ | $9.31 \cdot 10^3$ | $4.20 \cdot 10^2$ | 27.727 | $5.31 \cdot 10^2$ | 12.860 |
| MVO | $1.79 \cdot 10^4$ | $7.99 \cdot 10^3$ | n/a | n/a | $3.17 \cdot 10^2$ | 46.451 | $4.06 \cdot 10^2$ | **1.392** | $5.17 \cdot 10^2$ | 9.888 |
| PSO | $9.49 \cdot 10^4$ | $8.42 \cdot 10^2$ | n/a | n/a | $3.49 \cdot 10^2$ | 65.409 | $4.07 \cdot 10^2$ | 10.318 | $5.26 \cdot 10^2$ | 7.305 |
| WOA | $4.27 \cdot 10^7$ | $3.81 \cdot 10^6$ | n/a | n/a | $5.16 \cdot 10^3$ | $4.22 \cdot 10^2$ | $4.61 \cdot 10^2$ | 69.033 | $5.51 \cdot 10^2$ | 17.46 |
| SCA | $1.15 \cdot 10^8$ | $5.91 \cdot 10^7$ | n/a | n/a | $4.03 \cdot 10^3$ | $8.42 \cdot 10^2$ | $4.85 \cdot 10^2$ | 47.271 | $5.59 \cdot 10^2$ | 9.352 |
| AOA | $1.61 \cdot 10^5$ | $3.77 \cdot 10^4$ | n/a | n/a | $3.25 \cdot 10^2$ | 54.991 | $4.17 \cdot 10^2$ | 18.858 | $5.28 \cdot 10^2$ | 19.302 |
| HAOA | $\mathbf{1.30 \cdot 10^2}$ | **14.349** | n/a | n/a | $\mathbf{3.15 \cdot 10^2}$ | **28.129** | $4.07 \cdot 10^2$ | 2.369 | $\mathbf{4.98 \cdot 10^2}$ | 3.279 |

| Algorithm | F6 | | F7 | | F8 | | F9 | | F10 | |
|---|---|---|---|---|---|---|---|---|---|---|
| | Mean | STD | Mean | STD | Mean | STD | Mean | STD | Mean | STD |
| IHHO | $\mathbf{6.01 \cdot 10^2}$ | 0.082 | $7.49 \cdot 10^2$ | 10.041 | $8.11 \cdot 10^2$ | 6.526 | $1.13 \cdot 10^3$ | 85.42 | $1.69 \cdot 10^3$ | $1.31 \cdot 10^2$ |
| HHO | $6.38 \cdot 10^2$ | 12.320 | $7.96 \cdot 10^2$ | 18.921 | $8.29 \cdot 10^2$ | 5.700 | $1.44 \cdot 10^3$ | $1.24 \cdot 10^2$ | $2.03 \cdot 10^3$ | $3.42 \cdot 10^2$ |
| DE | $6.28 \cdot 10^2$ | 4.744 | $8.01 \cdot 10^2$ | 10.373 | $8.62 \cdot 10^2$ | 6.873 | $1.76 \cdot 10^3$ | $1.48 \cdot 10^2$ | $2.09 \cdot 10^3$ | $2.01 \cdot 10^2$ |
| GOA | $6.08 \cdot 10^2$ | 10.295 | $7.32 \cdot 10^2$ | 11.375 | $8.31 \cdot 10^2$ | 14.512 | $9.97 \cdot 10^2$ | 93.212 | $1.96 \cdot 10^3$ | $3.17 \cdot 10^2$ |
| GWO | $\mathbf{6.01 \cdot 10^2}$ | 1.909 | $7.35 \cdot 10^2$ | 16.343 | $8.16 \cdot 10^2$ | **5.053** | $9.14 \cdot 10^2$ | 12.11 | $1.76 \cdot 10^3$ | $3.10 \cdot 10^2$ |
| MFO | $6.02 \cdot 10^2$ | 2.411 | $7.46 \cdot 10^2$ | 22.655 | $8.29 \cdot 10^2$ | 13.786 | $1.23 \cdot 10^3$ | $2.76 \cdot 10^2$ | $2.02 \cdot 10^3$ | $3.27 \cdot 10^2$ |
| MVO | $6.03 \cdot 10^2$ | 4.365 | $7.30 \cdot 10^2$ | 11.278 | $8.25 \cdot 10^2$ | 12.216 | $\mathbf{9.00 \cdot 10^2}$ | 0.012 | $1.82 \cdot 10^3$ | $3.60 \cdot 10^2$ |
| PSO | $6.10 \cdot 10^2$ | 3.539 | $7.26 \cdot 10^2$ | **9.008** | $8.19 \cdot 10^2$ | 5.982 | $\mathbf{9.00 \cdot 10^2}$ | **0.003** | $\mathbf{1.50 \cdot 10^3}$ | $2.84 \cdot 10^2$ |
| WOA | $6.36 \cdot 10^2$ | 13.695 | $7.82 \cdot 10^2$ | 23.692 | $8.45 \cdot 10^2$ | 17.470 | $1.54 \cdot 10^3$ | $3.94 \cdot 10^2$ | $2.19 \cdot 10^3$ | $3.16 \cdot 10^2$ |
| SCA | $6.24 \cdot 10^2$ | 4.105 | $7.84 \cdot 10^2$ | 13.299 | $8.47 \cdot 10^2$ | 7.577 | $1.03 \cdot 10^3$ | 85.98 | $2.51 \cdot 10^3$ | $2.18 \cdot 10^2$ |
| AOA | $6.71 \cdot 10^2$ | 11.393 | $7.35 \cdot 10^2$ | 11.55 | $8.33 \cdot 10^2$ | 13.914 | $9.97 \cdot 10^2$ | 81.44 | $1.93 \cdot 10^3$ | $2.96 \cdot 10^2$ |
| HAOA | $\mathbf{6.01 \cdot 10^2}$ | **0.047** | $\mathbf{7.25 \cdot 10^2}$ | 11.393 | $\mathbf{8.06 \cdot 10^2}$ | 5.418 | $9.85 \cdot 10^2$ | 42.10 | $1.57 \cdot 10^3$ | $\mathbf{1.23 \cdot 10^2}$ |

| Algorithm | F11 | | F12 | | F13 | | F14 | | F15 | |
|---|---|---|---|---|---|---|---|---|---|---|
| | Mean | STD | Mean | STD | Mean | STD | Mean | STD | Mean | STD |
| IHHO | $1.13 \cdot 10^3$ | 13.523 | $4.25 \cdot 10^5$ | $3.05 \cdot 10^5$ | $4.42 \cdot 10^3$ | $2.18 \cdot 10^3$ | $\mathbf{1.42 \cdot 10^3}$ | **1.651** | $2.15 \cdot 10^3$ | $5.65 \cdot 10^2$ |
| HHO | $1.16 \cdot 10^3$ | 45.729 | $2.56 \cdot 10^6$ | $1.13 \cdot 10^6$ | $1.92 \cdot 10^4$ | $1.16 \cdot 10^4$ | $1.83 \cdot 10^3$ | $2.41 \cdot 10^2$ | $8.63 \cdot 10^3$ | $5.55 \cdot 10^2$ |
| DE | $1.14 \cdot 10^3$ | 36.317 | $9.15 \cdot 10^4$ | $6.58 \cdot 10^4$ | $\mathbf{1.35 \cdot 10^3}$ | 78.355 | $1.46 \cdot 10^3$ | 11.826 | $\mathbf{1.51 \cdot 10^3}$ | 18.454 |
| GOA | $1.17 \cdot 10^3$ | 58.009 | $2.24 \cdot 10^6$ | $1.15 \cdot 10^6$ | $1.65 \cdot 10^4$ | $1.13 \cdot 10^4$ | $2.93 \cdot 10^3$ | $1.15 \cdot 10^3$ | $6.48 \cdot 10^3$ | $4.32 \cdot 10^3$ |
| GWO | $1.34 \cdot 10^3$ | 183.524 | $1.31 \cdot 10^6$ | $1.54 \cdot 10^6$ | $1.26 \cdot 10^4$ | $7.82 \cdot 10^3$ | $3.19 \cdot 10^3$ | $1.82 \cdot 10^3$ | $5.63 \cdot 10^3$ | $3.16 \cdot 10^3$ |
| MFO | $1.23 \cdot 10^3$ | 107.133 | $2.23 \cdot 10^6$ | $4.81 \cdot 10^6$ | $1.61 \cdot 10^4$ | $1.39 \cdot 10^4$ | $8.42 \cdot 10^3$ | $5.42 \cdot 10^3$ | $1.25 \cdot 10^4$ | $1.02 \cdot 10^4$ |
| MVO | $1.14 \cdot 10^3$ | 27.331 | $1.52 \cdot 10^6$ | $1.41 \cdot 10^6$ | $9.89 \cdot 10^3$ | $2.55 \cdot 10^3$ | $2.15 \cdot 10^3$ | $1.03 \cdot 10^3$ | $4.05 \cdot 10^3$ | $2.45 \cdot 10^3$ |
| PSO | $\mathbf{1.12 \cdot 10^3}$ | 3.727 | $4.35 \cdot 10^4$ | $1.26 \cdot 10^4$ | $1.01 \cdot 10^4$ | $7.23 \cdot 10^3$ | $1.49 \cdot 10^3$ | 88.291 | $1.81 \cdot 10^3$ | $3.75 \cdot 10^2$ |
| WOA | $1.22 \cdot 10^3$ | 82.415 | $4.85 \cdot 10^6$ | $5.12 \cdot 10^6$ | $1.57 \cdot 10^4$ | $1.38 \cdot 10^4$ | $3.42 \cdot 10^3$ | $9.82 \cdot 10^2$ | $1.42 \cdot 10^4$ | $9.88 \cdot 10^3$ |
| SCA | $1.24 \cdot 10^3$ | 96.535 | $2.41 \cdot 10^7$ | $2.05 \cdot 10^7$ | $6.43 \cdot 10^4$ | $4.69 \cdot 10^4$ | $1.99 \cdot 10^3$ | $4.31 \cdot 10^2$ | $3.21 \cdot 10^3$ | $1.41 \cdot 10^3$ |
| AOA | $1.16 \cdot 10^3$ | 39.705 | $2.32 \cdot 10^6$ | $1.21 \cdot 10^6$ | $1.21 \cdot 10^4$ | $1.05 \cdot 10^4$ | $1.88 \cdot 10^3$ | $3.21 \cdot 10^2$ | $3.67 \cdot 10^3$ | $2.13 \cdot 10^3$ |
| HAOA | $\mathbf{1.12 \cdot 10^3}$ | **1.501** | $\mathbf{3.15 \cdot 10^4}$ | $\mathbf{2.24 \cdot 10^4}$ | $\mathbf{1.35 \cdot 10^3}$ | **20.495** | $1.46 \cdot 10^3$ | 21.354 | $\mathbf{1.51 \cdot 10^3}$ | **10.217** |

**Table 2.** *Cont.*

| Algorithm | F16 | | F17 | | F18 | | F19 | | F20 | |
| | **Mean** | **STD** | **Mean** | **STD** | **Mean** | **STD** | **Mean** | **STD** | **Mean** | **STD** |
| --- | --- | --- | --- | --- | --- | --- | --- | --- | --- | --- |
| IHHO | $1.73 \cdot 10^3$ | 59.44 | $1.73 \cdot 10^3$ | **7.519** | $4.79 \cdot 10^3$ | $1.68 \cdot 10^3$ | $\mathbf{1.95 \cdot 10^3}$ | 6.993 | $\mathbf{2.02 \cdot 10^3}$ | 19.561 |
| HHO | $1.89 \cdot 10^3$ | $1.47 \cdot 10^2$ | $1.79 \cdot 10^3$ | 65.751 | $2.02 \cdot 10^4$ | $1.41 \cdot 10^4$ | $1.71 \cdot 10^4$ | $1.21 \cdot 10^4$ | $2.23 \cdot 10^3$ | 86.017 |
| DE | $1.69 \cdot 10^3$ | **41.15** | $1.77 \cdot 10^3$ | 19.514 | $\mathbf{1.84 \cdot 10^3}$ | 23.298 | $2.75 \cdot 10^3$ | $8.35 \cdot 10^2$ | $2.05 \cdot 10^3$ | 23.711 |
| GOA | $1.78 \cdot 10^3$ | $1.76 \cdot 10^2$ | $1.83 \cdot 10^3$ | $1.21 \cdot 10^2$ | $1.63 \cdot 10^4$ | $1.31 \cdot 10^4$ | $3.25 \cdot 10^3$ | $1.95 \cdot 10^3$ | $2.15 \cdot 10^3$ | 74.824 |
| GWO | $1.79 \cdot 10^3$ | $1.11 \cdot 10^2$ | $1.77 \cdot 10^3$ | 38.759 | $2.55 \cdot 10^4$ | $1.84 \cdot 10^4$ | $2.75 \cdot 10^4$ | $2.38 \cdot 10^4$ | $2.09 \cdot 10^3$ | 73.994 |
| MFO | $1.85 \cdot 10^3$ | $15.23 \cdot 10^2$ | $1.78 \cdot 10^3$ | 65.311 | $2.21 \cdot 10^4$ | $1.39 \cdot 10^4$ | $7.81 \cdot 10^3$ | $6.15 \cdot 10^3$ | $2.13 \cdot 10^3$ | 72.321 |
| MVO | $1.80 \cdot 10^3$ | $1.44 \cdot 10^2$ | $1.80 \cdot 10^3$ | 46.126 | $2.03 \cdot 10^4$ | $1.25 \cdot 10^4$ | $4.63 \cdot 10^3$ | $2.62 \cdot 10^3$ | $2.12 \cdot 10^3$ | 86.303 |
| PSO | $\mathbf{1.65 \cdot 10^3}$ | 65.364 | $1.72 \cdot 10^3$ | 16.123 | $7.63 \cdot 10^3$ | $4.46 \cdot 10^3$ | $3.13 \cdot 10^3$ | $2.05 \cdot 10^3$ | $2.06 \cdot 10^3$ | 35.410 |
| WOA | $1.96 \cdot 10^3$ | $14.92 \cdot 10^2$ | $1.82 \cdot 10^3$ | 73.459 | $2.13 \cdot 10^4$ | $1.95 \cdot 10^2$ | $2.07 \cdot 10^5$ | $1.16 \cdot 10^5$ | $2.19 \cdot 10^3$ | $1.11 \cdot 10^2$ |
| SCA | $1.73 \cdot 10^3$ | 95.425 | $1.80 \cdot 10^3$ | 25.303 | $8.77 \cdot 10^4$ | $9.23 \cdot 10^2$ | $1.15 \cdot 10^4$ | $1.44 \cdot 10^3$ | $2.14 \cdot 10^3$ | 46.855 |
| AOA | $1.79 \cdot 10^3$ | $1.73 \cdot 10^2$ | $1.82 \cdot 10^3$ | $1.15 \cdot 10^2$ | $1.67 \cdot 10^4$ | $1.45 \cdot 10^4$ | $3.18 \cdot 10^3$ | $1.59 \cdot 10^3$ | $2.12 \cdot 10^3$ | 71.303 |
| HAOA | $1.71 \cdot 10^3$ | 86.348 | $\mathbf{1.72 \cdot 10^3}$ | 8.440 | $1.83 \cdot 10^3$ | **21.558** | $\mathbf{1.95 \cdot 10^3}$ | 8.716 | $\mathbf{2.02 \cdot 10^3}$ | **9.445** |

| Algorithm | F21 | | F22 | | F23 | | F24 | | F25 | |
| | **Mean** | **STD** | **Mean** | **STD** | **Mean** | **STD** | **Mean** | **STD** | **Mean** | **STD** |
| --- | --- | --- | --- | --- | --- | --- | --- | --- | --- | --- |
| IHHO | $\mathbf{2.21 \cdot 10^3}$ | **4.615** | $2.28 \cdot 10^3$ | 17.820 | $2.59 \cdot 10^3$ | 14.213 | $2.68 \cdot 10^3$ | $1.31 \cdot 10^2$ | $2.87 \cdot 10^3$ | 85.338 |
| HHO | $2.35 \cdot 10^3$ | 53.711 | $2.32 \cdot 10^3$ | 25.234 | $2.69 \cdot 10^3$ | 35.522 | $2.82 \cdot 10^3$ | 93.623 | $2.95 \cdot 10^3$ | 49.573 |
| DE | $2.25 \cdot 10^3$ | 78.104 | $2.29 \cdot 10^3$ | 17.513 | $2.63 \cdot 10^3$ | 15.163 | $\mathbf{2.66 \cdot 10^3}$ | 69.502 | $2.91 \cdot 10^3$ | **15.543** |
| GOA | $2.30 \cdot 10^3$ | 56.877 | $2.38 \cdot 10^3$ | $1.08 \cdot 10^2$ | $2.64 \cdot 10^3$ | 23.536 | $2.73 \cdot 10^3$ | 57.833 | $2.93 \cdot 10^3$ | 32.598 |
| GWO | $2.30 \cdot 10^3$ | 32.884 | $2.31 \cdot 10^3$ | 57.573 | $2.62 \cdot 10^3$ | 13.862 | $2.74 \cdot 10^3$ | 25.132 | $2.94 \cdot 10^3$ | 28.256 |
| MFO | $2.32 \cdot 10^3$ | 29.255 | $2.35 \cdot 10^3$ | 93.557 | $2.63 \cdot 10^3$ | 11.327 | $2.75 \cdot 10^3$ | 76.435 | $2.96 \cdot 10^3$ | 37.776 |
| MVO | $2.32 \cdot 10^3$ | 11.839 | $2.33 \cdot 10^3$ | $1.11 \cdot 10^2$ | $2.65 \cdot 10^3$ | **10.445** | $2.74 \cdot 10^3$ | 18.246 | $2.92 \cdot 10^3$ | 84.256 |
| PSO | $2.27 \cdot 10^3$ | 49.783 | $2.33 \cdot 10^3$ | $1.03 \cdot 10^2$ | $2.60 \cdot 10^3$ | 72.300 | $2.70 \cdot 10^3$ | 76.143 | $2.90 \cdot 10^3$ | 33.735 |
| WOA | $2.34 \cdot 10^3$ | 60.021 | $2.48 \cdot 10^3$ | $2.45 \cdot 10^2$ | $2.66 \cdot 10^3$ | 29.838 | $2.77 \cdot 10^3$ | 85.902 | $2.98 \cdot 10^3$ | $1.03 \cdot 10^2$ |
| SCA | $2.29 \cdot 10^3$ | 65.229 | $2.41 \cdot 10^3$ | 66.636 | $2.67 \cdot 10^3$ | 45.449 | $2.78 \cdot 10^3$ | **11.548** | $2.98 \cdot 10^3$ | 37.291 |
| AOA | $2.29 \cdot 10^3$ | 34.701 | $2.36 \cdot 10^3$ | $1.10 \cdot 10^2$ | $2.62 \cdot 10^3$ | 17.452 | $2.72 \cdot 10^3$ | $1.05 \cdot 10^2$ | $2.93 \cdot 10^3$ | 47.019 |
| HAOA | $\mathbf{2.21 \cdot 10^3}$ | 8.551 | $\mathbf{2.25 \cdot 10^3}$ | **13.041** | $\mathbf{2.56 \cdot 10^3}$ | 21.928 | $2.67 \cdot 10^3$ | $1.71 \cdot 10^2$ | $\mathbf{2.80 \cdot 10^3}$ | 95.426 |

| Algorithm | F26 | | F27 | | F28 | | F29 | | F30 | |
| | **Mean** | **STD** | **Mean** | **STD** | **Mean** | **STD** | **Mean** | **STD** | **Mean** | **STD** |
| --- | --- | --- | --- | --- | --- | --- | --- | --- | --- | --- |
| IHHO | $2.93 \cdot 10^3$ | $1.66 \cdot 10^2$ | $3.19 \cdot 10^3$ | 33.657 | $3.30 \cdot 10^3$ | 48.694 | $\mathbf{3.20 \cdot 10^3}$ | 28.982 | $2.30 \cdot 10^4$ | $1.45 \cdot 10^4$ |
| HHO | $3.62 \cdot 10^3$ | $5.39 \cdot 10^2$ | $3.18 \cdot 10^3$ | 51.306 | $3.41 \cdot 10^3$ | $1.02 \cdot 10^2$ | $3.39 \cdot 10^3$ | 85.653 | $1.43 \cdot 10^6$ | $1.31 \cdot 10^6$ |
| DE | $2.95 \cdot 10^3$ | 95.929 | $\mathbf{3.07 \cdot 10^3}$ | **2.558** | $3.28 \cdot 10^3$ | **27.035** | $3.21 \cdot 10^3$ | 35.216 | $3.65 \cdot 10^5$ | $2.31 \cdot 10^5$ |
| GOA | $3.01 \cdot 10^3$ | $3.65 \cdot 10^2$ | $3.11 \cdot 10^3$ | 25.326 | $3.31 \cdot 10^3$ | $1.53 \cdot 10^2$ | $3.27 \cdot 10^3$ | 75.411 | $5.29 \cdot 10^5$ | $3.89 \cdot 10^5$ |
| GWO | $3.36 \cdot 10^3$ | $5.05 \cdot 10^2$ | $3.10 \cdot 10^3$ | 13.541 | $3.42 \cdot 10^3$ | $1.33 \cdot 10^2$ | $3.22 \cdot 10^3$ | 49.822 | $6.17 \cdot 10^5$ | $4.88 \cdot 10^5$ |
| MFO | $3.05 \cdot 10^3$ | $1.13 \cdot 10^2$ | $3.09 \cdot 10^3$ | 5.722 | $3.21 \cdot 10^3$ | 93.459 | $3.26 \cdot 10^3$ | 55.593 | $6.36 \cdot 10^5$ | $5.93 \cdot 10^5$ |
| MVO | $3.15 \cdot 10^3$ | $2.77 \cdot 10^2$ | $3.10 \cdot 10^3$ | 21.875 | $3.36 \cdot 10^3$ | $1.23 \cdot 10^2$ | $3.26 \cdot 10^3$ | 75.139 | $4.62 \cdot 10^5$ | $4.07 \cdot 10^5$ |
| PSO | $2.95 \cdot 10^3$ | $2.55 \cdot 10^2$ | $3.12 \cdot 10^3$ | 31.830 | $3.32 \cdot 10^3$ | $1.35 \cdot 10^2$ | $3.21 \cdot 10^3$ | 62.374 | $1.13 \cdot 10^6$ | $1.09 \cdot 10^6$ |
| WOA | $3.37 \cdot 10^3$ | $2.92 \cdot 10^2$ | $3.17 \cdot 10^3$ | 48.124 | $3.46 \cdot 10^3$ | $1.65 \cdot 10^2$ | $3.46 \cdot 10^3$ | $1.21 \cdot 10^2$ | $1.29 \cdot 10^6$ | $7.53 \cdot 10^5$ |
| SCA | $3.15 \cdot 10^3$ | $1.82 \cdot 10^2$ | $3.13 \cdot 10^3$ | 13.152 | $3.38 \cdot 10^3$ | 89.259 | $3.25 \cdot 10^3$ | 48.339 | $1.49 \cdot 10^6$ | $9.77 \cdot 10^5$ |
| AOA | $3.02 \cdot 10^3$ | $2.03 \cdot 10^2$ | $3.10 \cdot 10^3$ | 27.015 | $3.32 \cdot 10^3$ | $1.17 \cdot 10^2$ | $3.26 \cdot 10^3$ | 31.117 | $4.71 \cdot 10^5$ | $4.02 \cdot 10^5$ |
| HAOA | $\mathbf{2.84 \cdot 10^3}$ | $2.46 \cdot 10^2$ | $3.09 \cdot 10^3$ | 48.691 | $\mathbf{3.11 \cdot 10^3}$ | $2.53 \cdot 10^2$ | $\mathbf{3.20 \cdot 10^3}$ | 27.909 | $\mathbf{2.21 \cdot 10^4}$ | $\mathbf{1.42 \cdot 10^4}$ |

Observing Table 3, the HAOA undoubtedly performs better than any of the other 11 algorithms taken into account for comparative analysis. As expected, the second best approach is IHHO, while the original AOA and SCA take the ranks of 6 and 11, respectively. Additionally, the calculated Friedman statistics $\chi_r^2$ is 21.672, and as such, it is greater than the $\chi^2$ critical value with 11 degrees of freedom ($1.9675 \times 10^1$) at the threshold level of $\alpha = 0.05$. The conclusion of this analysis is that the null hypothesis (*H*0) can be rejected, implying that the HAOA achieved results which are substantially better than other algorithms.

The convergence speed visual difference between the proposed HAOA and AOA, SCA, as well as between the other three best-performing metaheuristics, IHHO, DE and PSO for *F*4, *F*6, *F*11, *F*17, *F*22 and *F*28 instances, is shown in Figure 5. From the sample functions convergence

graphs, it can be observed that the HAOA converges on average faster than other methods, which is particularly emphasized in cases of *F*4, *F*6 and *F*11 instance. It can also be seen that the results' quality generated by HAOA is much higher than its base algorithms, AOA and SCA.

**Table 3.** Friedman test ranks for the compared algorithms over 29 CEC2017 functions.

| Function | IHHO | HHO | DE | GOA | GWO | MFO | MVO | PSO | WOA | SCA | AOA | HAOA |
|----------|------|-----|-----|------|------|------|------|------|--------|--------|--------|--------|
| *F*1 | 2 | 7 | 11 | 5 | 9 | 8 | 3 | 4 | 10 | 12 | 6 | 1 |
| *F*3 | 1.5 | 7 | 10 | 3.5 | 8 | 12 | 3.5 | 6 | 11 | 9 | 5 | 1 |
| *F*4 | 1 | 10 | 9 | 6 | 5 | 8 | 3 | 4 | 11 | 12 | 7 | 2 |
| *F*5 | 2 | 9 | 11 | 5 | 4 | 8 | 3 | 6 | 10 | 12 | 7 | 1 |
| *F*6 | 1.5 | 11 | 9 | 6 | 3 | 4 | 5 | 7 | 10 | 8 | 12 | 1.5 |
| *F*7 | 8 | 11 | 12 | 4 | 5.5 | 7 | 3 | 2 | 9 | 10 | 5.5 | 1 |
| *F*8 | 2 | 6.5 | 12 | 8 | 3 | 6.5 | 5 | 4 | 10 | 11 | 9 | 1 |
| *F*9 | 8 | 10 | 12 | 5.5 | 3 | 9 | 1.5 | 1.5 | 11 | 7 | 5.5 | 4 |
| *F*10 | 3 | 9 | 10 | 7 | 4 | 8 | 5 | 1 | 11 | 12 | 6 | 2 |
| *F*11 | 3 | 6.5 | 4.5 | 8 | 12 | 10 | 4.5 | 1.5 | 9 | 11 | 6.5 | 1.5 |
| *F*12 | 4 | 10 | 3 | 8 | 5 | 7 | 6 | 2 | 11 | 12 | 9 | 1 |
| *F*13 | 3 | 11 | 1.5 | 10 | 7 | 9 | 4 | 5 | 8 | 12 | 6 | 1.5 |
| *F*14 | 1 | 5 | 3 | 9 | 10 | 12 | 8 | 4 | 11 | 7 | 6 | 2 |
| *F*15 | 4 | 10 | 1.5 | 9 | 8 | 11 | 7 | 3 | 12 | 5 | 6 | 1.5 |
| *F*16 | 5.5 | 11 | 3 | 1 | 7.5 | 10 | 9 | 2 | 12 | 5.5 | 7.5 | 4 |
| *F*17 | 3 | 7 | 4.5 | 12 | 4.5 | 6 | 8.5 | 2 | 10.5 | 8.5 | 10.5 | 1 |
| *F*18 | 3 | 7 | 1 | 5 | 11 | 10 | 8 | 4 | 9 | 12 | 6 | 2 |
| *F*19 | 1.5 | 10 | 3 | 6 | 11 | 8 | 7 | 4 | 12 | 9 | 5 | 1.5 |
| *F*20 | 2 | 12 | 3 | 10 | 5 | 8 | 6.5 | 4 | 11 | 9 | 6.5 | 1 |
| *F*21 | 1.5 | 12 | 3 | 7.5 | 7.5 | 9.5 | 9.5 | 4 | 11 | 5.5 | 5.5 | 1.5 |
| *F*22 | 2 | 5 | 3 | 10 | 4 | 8 | 6.5 | 6.5 | 12 | 11 | 9 | 1 |
| *F*23 | 2 | 12 | 6.5 | 8 | 4.5 | 6.5 | 9 | 3 | 10 | 11 | 4.5 | 1 |
| *F*24 | 3 | 12 | 1 | 6 | 7.5 | 9 | 7.5 | 4 | 10 | 11 | 5 | 2 |
| *F*25 | 2 | 9 | 4 | 6.5 | 8 | 10 | 5 | 3 | 11.5 | 11.5 | 6.5 | 1 |
| *F*26 | 2 | 12 | 3.5 | 5 | 10 | 7 | 8.5 | 3.5 | 11 | 8.5 | 6 | 1 |
| *F*27 | 12 | 11 | 1 | 7 | 5 | 3 | 5 | 8 | 10 | 9 | 5 | 2 |
| *F*28 | 4 | 10 | 3 | 5 | 11 | 2 | 8 | 6.5 | 12 | 9 | 6.5 | 1 |
| *F*29 | 1.5 | 11 | 3.5 | 10 | 5 | 8 | 8 | 3.5 | 12 | 6 | 8 | 1.5 |
| *F*30 | 2 | 11 | 3 | 6 | 7 | 8 | 4 | 9 | 10 | 12 | 5 | 1 |
| Average Ranking | 3.138 | 9.483 | 5.362 | 6.862 | 6.724 | 8.017 | 5.914 | 4.069 | 10.621 | 9.603 | 6.655 | 1.552 |
| Rank | 2 | 10 | 4 | 8 | 7 | 9 | 5 | 3 | 12 | 11 | 6 | 1 |

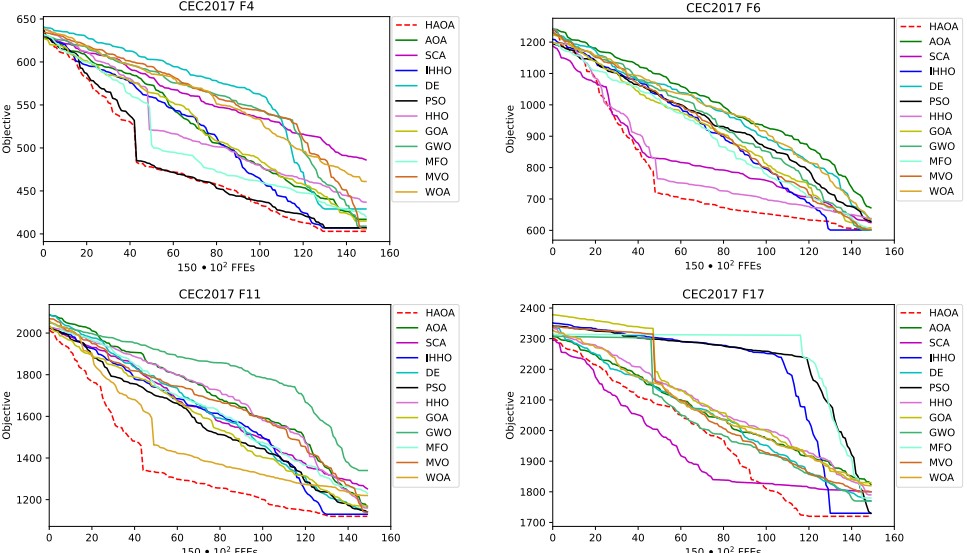

**Figure 5.** *Cont.*

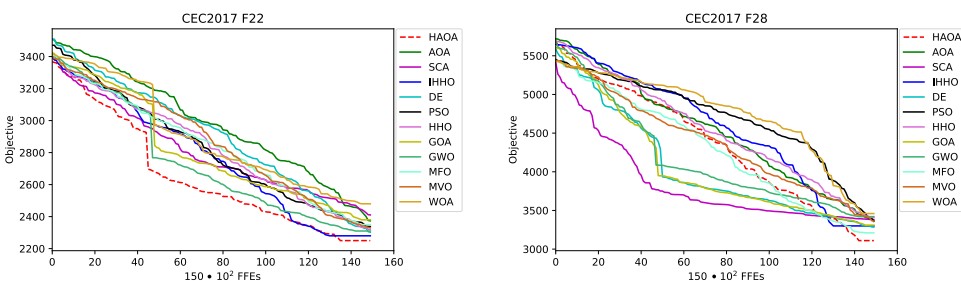

**Figure 5.** CEC 2017 experiments convergence speed graphs for *F*4, *F*6, *F*11, *F*17, *F*22 and *F*28 benchmarks for some approaches

## 5. The COVID-19 X-ray Images Classification Findings

This section first provides an overview of datasets used in experiments, which is followed by details of experimental setup and comparative analysis. This section concludes with the validation of experimental findings.

### 5.1. Dataset Description

The majority of images for the dataset employed in this research is taken from the COVID-19 radiography database, which can be retrieved from the following URL: https://www.kaggle.com/datasets/tawsifurrahman/covid19-radiography-database (accessed on 25 October 2022). The lung opacity images are excluded, and the remaining three groups are taken for experiments. The images are categorized as follows: normal (class 0), COVID-19 (class 1) and viral pneumonia (class 2). The retrieved dataset includes sets of 3616, 10,192 and 1345 images for COVID-19, normal and viral pneumonia classes, respectively. The COVID-19 radiography database images were also employed in other research [33,34]. Random sample images from the COVID-19 radiography database are shown in Figure 6.

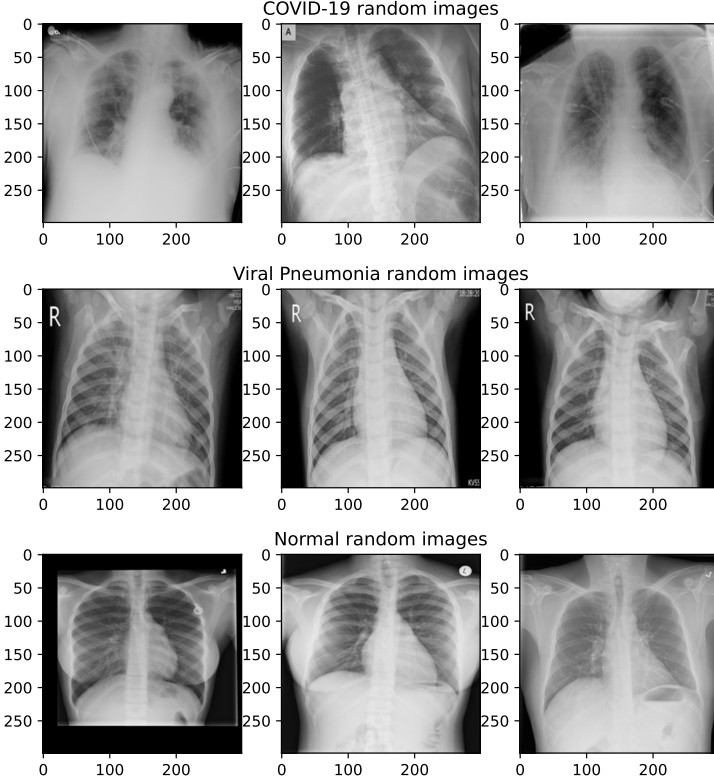

**Figure 6.** Sample X-ray images for normal, COVID-19 and pneumonia taken from the COVID-19 radiography database.

According to the above description, the COVID-19 radiography repository is unbalanced, containing a majority of non-infected (normal) lung images. Since the area of the proposed research is not related to addressing imbalanced datasets, the utilized dataset is balanced so that each class has 4000 images. The balancing is performed in the following way: a random subset of 4000 normal images is taken from the original set, and the COVID-19 images are supplemented up to 4000 by taking some X-ray COVID-19 images from the Augmented COVID-19 X-ray Images Dataset [136] and by generated dedicated augmented images for this research, while 2655 viral pneumonia additional figures are generated by performing geometric augmentation of the original ones.

Samples for COVID-19 and viral pneumonia generated augmented images for the purpose of this research are shown in Figure 7, while the classes distribution of the original (imbalanced) COVID-19 radiography repository and artificially generated (balanced) dataset used in this research are presented in Figure 8.

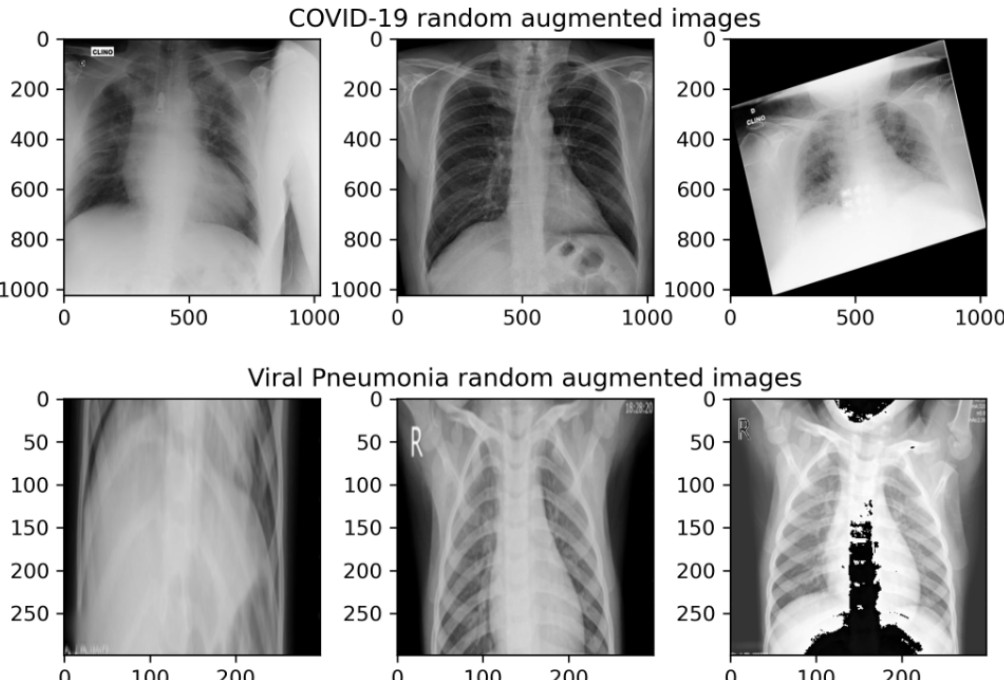

**Figure 7.** Sample X-ray images for augmented COVID-19 and viral pneumonia classes generated for this research.

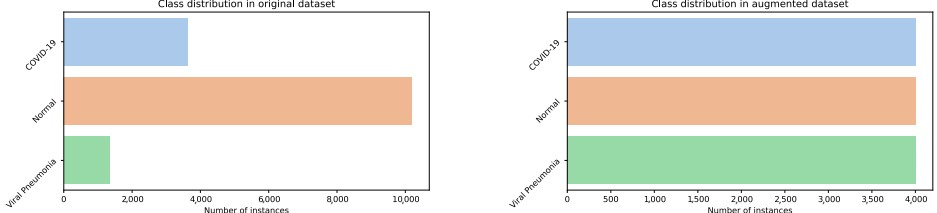

**Figure 8.** Distribution of classes in original and augmented COVID-19 radiography dataset.

### 5.2. Experimental Setup, Comparative Analysis Reports and Discussion

The research shown in this paper uses a similar experimental setup as in [137], where chimp optimization algorithm (ChOA) was used to determine initial weights and biases for extreme learning machine (ELM) which is used to classify features extracted from simple CNN for smaller COVID-19 X-ray images dataset. Converserly to [137], this research uses even simpler CNN for feature extraction and XGBoost classifier and a much larger COVID-19 X-ray image set.

To establish the performance of the CNN–XGBoost–HAOA proposed approach, a comparative analysis with other evolved XGBoost structures by using eight other metaheuristics is performed. The comparative analysis considered the following metaheuristics: basic AOA, SCA, IHHO [125], HHO [126], PSO [83], DE [127], teaching–learning-based optimization (TLB) [138] and ChOA [139]. Therefore, besides AOA and SCA as baseline methods for HAOA, the algorithm set for comparative analysis also included three best-performing approaches in CEC2017 simulations (Section 4), as well as a few other metaheuristics. It is noted that in the results' tables, as well as figures, for readability reasons, the XGBoost is abbreviated as XG.

All methods were tested under the same experimental condition. The COVID-19 X-ray dataset is first split by using a stratified *train_test_split* method in proportions of 70%, 30% for the train and test sets, respectively. Afterwards, simple CNN, as shown in Figure 2 is trained on the training set and tested on the testing set in the first phase of the proposed methodology, as described in Section 3.3. Afterwards, outputs from the CNN's flatten layer were extracted separately for training and testing sets, and those sets were used as inputs for XGBoost, which is then tuned by metaheurisitcs.

All methods were tested with 20 solutions in the population ($N = 20$), and XGBoost structures were tuned throughout 30 iterations ($T = 30$) and 15 separated runs ($R = 15$). The classification error rate for the training set is used as an objective function. After completing one runtime, the best-performing XGBoost model was validated against the testing set, and this was reported as the best solution in the run. Afterwards, the best, mean, worst, median, standard deviation and variance metrics of the best solutions' testing set objective (error rate) for each metaheuristics over 15 runs are captured and reported in Table 4.

**Table 4.** The best, worst, mean, median, standard deviation and variance of classification error rate for 15 independent runs—CNN–XGBoost–HAOA vs. others.

| Method | Best | Worst | Mean | Median | Std | Var |
|---|---|---|---|---|---|---|
| CNN–XG–HAOA | $\mathbf{6.11 \cdot 10^{-3}}$ | $\mathbf{8.61 \cdot 10^{-3}}$ | $\mathbf{7.81 \cdot 10^{-3}}$ | $\mathbf{7.92 \cdot 10^{-3}}$ | $6.39 \cdot 10^{-4}$ | $4.08 \cdot 10^{-7}$ |
| CNN–XG–AOA | $7.78 \cdot 10^{-3}$ | $9.45 \cdot 10^{-3}$ | $8.42 \cdot 10^{-3}$ | $8.34 \cdot 10^{-3}$ | $4.66 \cdot 10^{-4}$ | $2.17 \cdot 10^{-7}$ |
| CNN–XG–SCA | $7.78 \cdot 10^{-3}$ | $9.17 \cdot 10^{-3}$ | $8.53 \cdot 10^{-3}$ | $8.61 \cdot 10^{-3}$ | $4.31 \cdot 10^{-4}$ | $1.86 \cdot 10^{-7}$ |
| CNN–XG–IHHO | $7.22 \cdot 10^{-3}$ | $8.89 \cdot 10^{-3}$ | $8.25 \cdot 10^{-3}$ | $8.34 \cdot 10^{-3}$ | $4.66 \cdot 10^{-4}$ | $2.17 \cdot 10^{-7}$ |
| CNN–XG–HHO | $7.78 \cdot 10^{-3}$ | $9.45 \cdot 10^{-3}$ | $8.70 \cdot 10^{-3}$ | $8.61 \cdot 10^{-3}$ | $4.66 \cdot 10^{-4}$ | $2.17 \cdot 10^{-7}$ |
| CNN–XG–PSO | $8.06 \cdot 10^{-3}$ | $8.89 \cdot 10^{-3}$ | $8.56 \cdot 10^{-3}$ | $8.61 \cdot 10^{-3}$ | $\mathbf{2.99 \cdot 10^{-4}}$ | $\mathbf{8.96 \cdot 10^{-8}}$ |
| CNN–XG–DE | $7.50 \cdot 10^{-3}$ | $9.17 \cdot 10^{-3}$ | $8.31 \cdot 10^{-3}$ | $8.47 \cdot 10^{-3}$ | $5.19 \cdot 10^{-4}$ | $2.69 \cdot 10^{-7}$ |
| CNN–XG–TLB | $8.06 \cdot 10^{-3}$ | $9.72 \cdot 10^{-3}$ | $8.92 \cdot 10^{-3}$ | $9.03 \cdot 10^{-3}$ | $5.04 \cdot 10^{-4}$ | $2.54 \cdot 10^{-7}$ |
| CNN–XG–ChOA | $8.34 \cdot 10^{-3}$ | $9.72 \cdot 10^{-3}$ | $9.09 \cdot 10^{-3}$ | $9.17 \cdot 10^{-3}$ | $3.94 \cdot 10^{-4}$ | $1.55 \cdot 10^{-7}$ |

As shown in Table 4, the proposed CNN–XGBoost–HAOA approach obtained predominant results by achieving the best values for best, worst, mean and median metrics, while CNN–XGBoost–IHHO finished second. Both baseline methods, CNN–XGBoost–AOA and CNN–XGBoost–SCA, obtained average results, and were far behind the hybrid approach proposed in this research. The best values for *std* and *var* metrics were obtained by CNN–XGBoost–PSO, indicating that this approach delivers the most stable results (consistent, but even the best score obtained by the CNN–XGBoost–PSO was behind the mean result of the proposed algorithm).

Additionally, detailed metrics in terms of precision, recall and F1-score per classes along with accuracy and micro weighted metrics are also captured for the best-performing metaheuristics solution and for the CNN structure introduced in Section 3.3 and shown in Figure 2, which was used for feature extraction. These results are presented in Table 5. For clarity reasons, the prefix 'CNN-XGBoost' is omitted in the header of the detailed results table.

**Table 5.** Detailed metrics for best-performing solution and baseline CNN.

| Methods | HAOA | AOA | SCA | IHHO | HHO | PSO | DE | TLB | ChOA | CNN |
|---|---|---|---|---|---|---|---|---|---|---|
| Acc. (%) | **99.3887** | 99.2220 | 99.2220 | 99.2776 | 99.2220 | 99.1942 | 99.2498 | 99.1942 | 99.1664 | 97.5000 |
| Precision 0 | 0.990826 | 0.989975 | 0.989983 | **0.993272** | 0.992450 | 0.989958 | 0.991632 | 0.989158 | 0.991604 | 0.958983 |
| Precision 1 | **0.994176** | 0.991694 | 0.990041 | 0.987603 | 0.989247 | 0.990864 | 0.990871 | 0.990033 | 0.986777 | 0.978796 |
| Precision 2 | 0.996661 | 0.994992 | 0.996656 | **0.997500** | 0.994992 | 0.995000 | 0.994996 | 0.996656 | 0.996661 | 0.987521 |
| M.Avg. Pr. | **0.993889** | 0.992221 | 0.992227 | 0.992792 | 0.992229 | 0.991941 | 0.992500 | 0.991950 | 0.991681 | 0.975100 |
| Recall 0 | **0.990826** | 0.988324 | 0.989158 | 0.984987 | 0.986656 | 0.986656 | 0.988324 | 0.989158 | 0.984987 | 0.974167 |
| Recall 1 | 0.995833 | 0.995000 | 0.994167 | 0.995833 | **0.996667** | 0.994167 | 0.995000 | 0.993333 | 0.995000 | 0.961667 |
| Recall 2 | 0.995000 | 0.993333 | 0.993333 | **0.997500** | 0.993333 | 0.995000 | 0.994167 | 0.993333 | 0.995000 | 0.989167 |
| M.Avg. Rec | **0.993887** | 0.992220 | 0.992220 | 0.992776 | 0.992220 | 0.991942 | 0.992498 | 0.991942 | 0.991664 | 0.975000 |
| F1-score 0 | **0.990826** | 0.989149 | 0.989570 | 0.989112 | 0.989544 | 0.988304 | 0.989975 | 0.989158 | 0.988285 | 0.966515 |
| F1-score 1 | **0.995004** | 0.993344 | 0.992100 | 0.991701 | 0.992943 | 0.992512 | 0.992931 | 0.991681 | 0.990871 | 0.970156 |
| F1-score 2 | 0.995830 | 0.994162 | 0.994992 | **0.997500** | 0.994162 | 0.995000 | 0.994581 | 0.994992 | 0.995830 | 0.988343 |
| M.Avg. F1 | **0.993887** | 0.992219 | 0.992221 | 0.992772 | 0.992217 | 0.991940 | 0.992497 | 0.991944 | 0.991663 | 0.975005 |

From Table 5, the first thing that is interesting to emphasize is that all metaheuristics performed much better than the CNN used for feature extraction. Therefore, the XGBoost showed better performance for the classification of extracted features than standard fully connected layers of the CNN. When analyzing the performance level of metaheuristics-based models, the proposed CNN–XGBoost–HAOA obtained the best results for eight out of thirteen metrics, while the CNN–XBGBoost–IHHO finished second by obtaining the best scores for four metrics. The highest accuracy of almost 99.4% was also achieved by the CNN–XGBoost–HAOA method. Again, it is worth noting that the hybrid algorithm significantly outperformed both baseline metaherustics (AOA and SCA) in all observed categories.

Additionally, the results of the best-performing solutions in terms of true positives (TP), true negatives (TN), false positives (FP) and false negatives (FN), true positive rate (TPR, sensitivity or recall), true negative rate (TPR, specificity), positive predicted values (PPV, precision), negative predictive values (NPV), false positive rate (FPR), false negative rate (FNR) and false discovery rate (FDR) are shown in Table 6.

The set of hyperparameters' values for best evolved XGBoost structures is shown in Table 7.

The convergence speed graph for the best quality CNN-XGBoost metaheuristics solutions along with diversity over 15 runs are visualized in Figure 9. It can be noticed that the proposed CNN–XGboost–HAOA establishes fastest convergence and how it performs a search with huge improvements—for some iterations, it becomes stuck in sub-optimal regions; however, it eventually manages to get away and converge toward optimum.

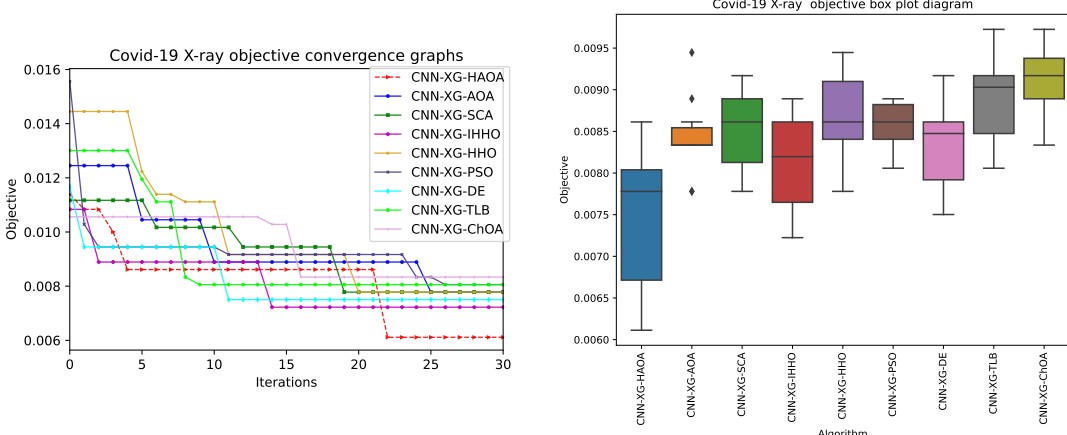

**Figure 9.** Convergence speed graph for best-quality CNN-XGBoost metaheuristics solution and solutions' diversity over 15 runs.

**Table 6.** Results of the best-performing solutions of each algorithm.

| | Class | FP | FN | TP | TN | TPR | TNR | PPV | NPV | FPR | FNR | FDR |
|---|---|---|---|---|---|---|---|---|---|---|---|---|
| CNN–XG–HAOA | Normal | 11 | 11 | 1188 | 2389 | 0.991 | 0.995 | 0.991 | 0.995 | 0.005 | 0.009 | 0.009 |
| | COVID-19 | 7 | 5 | 1195 | 2392 | 0.996 | 0.997 | 0.994 | 0.998 | 0.003 | 0.004 | 0.006 |
| | Pneumonia | 4 | 6 | 1194 | 2395 | 0.995 | 0.998 | 0.997 | 0.998 | 0.002 | 0.005 | 0.003 |
| CNN–XG–AOA | Normal | 12 | 14 | 1185 | 2388 | 0.988 | 0.995 | 0.99 | 0.994 | 0.005 | 0.012 | 0.010 |
| | COVID-19 | 10 | 6 | 1194 | 2389 | 0.995 | 0.996 | 0.992 | 0.997 | 0.004 | 0.005 | 0.008 |
| | Pneumonia | 6 | 8 | 1192 | 2393 | 0.993 | 0.997 | 0.995 | 0.997 | 0.003 | 0.007 | 0.005 |
| CNN–XG–SCA | Normal | 12 | 13 | 1186 | 2388 | 0.989 | 0.995 | 0.99 | 0.995 | 0.005 | 0.011 | 0.010 |
| | COVID-19 | 12 | 7 | 1193 | 2387 | 0.994 | 0.995 | 0.99 | 0.997 | 0.005 | 0.006 | 0.010 |
| | Pneumonia | 4 | 8 | 1192 | 2395 | 0.993 | 0.998 | 0.997 | 0.997 | 0.002 | 0.007 | 0.003 |
| CNN–XG–IHHO | Normal | 8 | 18 | 1181 | 2392 | 0.985 | 0.997 | 0.993 | 0.993 | 0.003 | 0.015 | 0.007 |
| | COVID-19 | 15 | 5 | 1195 | 2384 | 0.996 | 0.994 | 0.988 | 0.998 | 0.006 | 0.004 | 0.012 |
| | Pneumonia | 3 | 3 | 1197 | 2396 | 0.998 | 0.999 | 0.998 | 0.999 | 0.001 | 0.002 | 0.002 |
| CNN–XG–HHO | Normal | 9 | 16 | 1183 | 2391 | 0.987 | 0.996 | 0.992 | 0.993 | 0.004 | 0.013 | 0.008 |
| | COVID-19 | 13 | 4 | 1196 | 2386 | 0.997 | 0.995 | 0.989 | 0.998 | 0.005 | 0.003 | 0.011 |
| | Pneumonia | 6 | 8 | 1192 | 2393 | 0.993 | 0.997 | 0.995 | 0.997 | 0.003 | 0.007 | 0.005 |
| CNN–XG–PSO | Normal | 12 | 16 | 1183 | 2388 | 0.987 | 0.995 | 0.99 | 0.993 | 0.005 | 0.013 | 0.010 |
| | COVID-19 | 11 | 7 | 1193 | 2388 | 0.994 | 0.995 | 0.991 | 0.997 | 0.005 | 0.006 | 0.009 |
| | Pneumonia | 6 | 6 | 1194 | 2393 | 0.995 | 0.997 | 0.995 | 0.997 | 0.003 | 0.005 | 0.005 |
| CNN–XG–DE | Normal | 10 | 14 | 1185 | 2390 | 0.988 | 0.996 | 0.992 | 0.994 | 0.004 | 0.012 | 0.008 |
| | COVID-19 | 11 | 6 | 1194 | 2388 | 0.995 | 0.995 | 0.991 | 0.997 | 0.005 | 0.005 | 0.009 |
| | Pneumonia | 6 | 7 | 1193 | 2393 | 0.994 | 0.997 | 0.995 | 0.997 | 0.003 | 0.006 | 0.005 |
| CNN–XG–TLB | Normal | 13 | 13 | 1186 | 2387 | 0.989 | 0.995 | 0.989 | 0.995 | 0.005 | 0.011 | 0.011 |
| | COVID-19 | 12 | 8 | 1192 | 2387 | 0.993 | 0.995 | 0.99 | 0.997 | 0.005 | 0.007 | 0.010 |
| | Pneumonia | 4 | 8 | 1192 | 2395 | 0.993 | 0.998 | 0.997 | 0.997 | 0.002 | 0.007 | 0.003 |
| CNN–XG–ChOA | Normal | 10 | 18 | 1181 | 2390 | 0.985 | 0.996 | 0.992 | 0.993 | 0.004 | 0.015 | 0.008 |
| | COVID-19 | 16 | 6 | 1194 | 2383 | 0.995 | 0.993 | 0.987 | 0.997 | 0.007 | 0.005 | 0.013 |
| | Pneumonia | 4 | 6 | 1194 | 2395 | 0.995 | 0.998 | 0.997 | 0.998 | 0.002 | 0.005 | 0.003 |

**Table 7.** Best solutions' XGBoost hyperparameters value.

| | l.r. (μ) | max_child_weight | Subsample | collsample_bytree | max_depth | Gamma |
|---|---|---|---|---|---|---|
| CNN–XG–HAOA | 0.900000 | 1.590710 | 1.000000 | 0.282676 | 10 | 0.000000 |
| CNN–XG–AOA | 0.900000 | 1.042019 | 0.761057 | 0.437239 | 6 | 0.394364 |
| CNN–XG–SCA | 0.662726 | 1.000000 | 1.000000 | 0.696119 | 8 | 0.000000 |
| CNN–XG–IHHO | 0.691881 | 1.453556 | 1.000000 | 0.222989 | 8 | 0.000000 |
| CNN–XG–HHO | 0.884780 | 1.000000 | 0.887372 | 0.391540 | 10 | 0.088580 |
| CNN–XG–PSO | 0.889012 | 7.076648 | 1.000000 | 0.610577 | 7 | 0.038919 |
| CNN–XG–DE | 0.900000 | 6.335094 | 1.000000 | 0.628815 | 10 | 0.000000 |
| CNN–XG–TLB | 0.900000 | 1.173684 | 0.904359 | 0.390759 | 10 | 0.018695 |
| CNN–XG–ChOA | 0.900000 | 1.000000 | 1.000000 | 0.297912 | 10 | 0.746814 |

Finally, to better visualize the performance of CNN–XGBoost–HAOA, the confusion matrix, receiver operating characteristics (ROC) and precision–recall (PR) curves along with ROC all vs. rest (OvR) for the best solution are visualized in Figure 10.

To validate the findings from COVID-19 X-ray simulations, the best values for each of the 15 independent runs are taken for comparison for every metaheuristics method, and all algorithms were compared by using a non-parametric test. However, prior to rendering the decision of using a non-parametric test, the safe use of parametric tests, which includes the independence, normality, and homoscedasticity of the data variances, was checked [140]. The condition of independence is satisfied because each run starts with different pseudo-random number seeds. The homoscedasticity is validated by performing

Levene's test [141], and the *p*-value of 0.67 is obtained in all cases, rendering the conclusion that the homoscedasticity is satisfied.

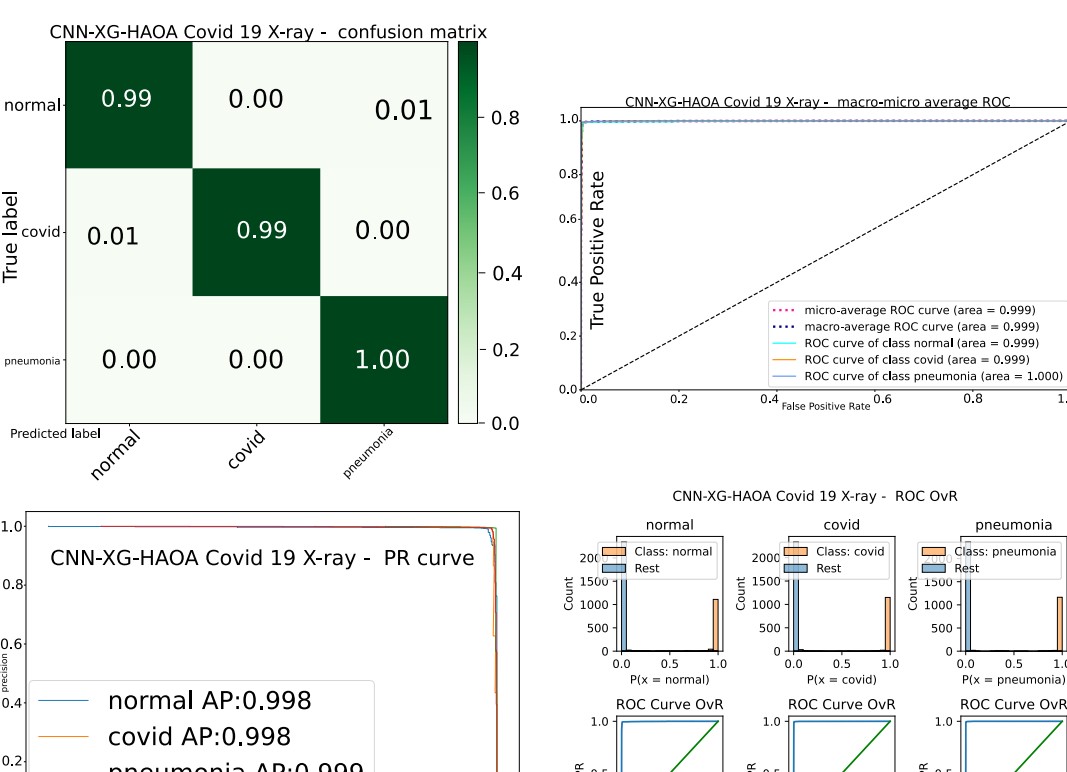

**Figure 10.** Convergence speed graph for best-quality CNN–XGBoost metaheuristics solution and solutions' diversity over 15 runs.

Finally, the Shapiro–Wilk test for single-problem analysis [142] was conducted to check whether or not the results from independent runs originated from normal distribution in the following way: results series that include the best solution in each run are constructed for each metaheuristics, and Shapiro–Wilk *p*-values are calculated for each method separately. The obtained *p*-value for each algorithm was lower than 0.05, allowing the conclusion that the *H*0 hypothesis was rejected for both $alpha = 0.1$ and $alpha = 0.05$. This means that the results are not originated from the normal distribution. The results of the Shapiro test are briefly summarized in Table 8.

Therefore, since the normality condition was not satisfied, it was proceeded with a nonparametric Wilcoxon signed-rank test [143] with the same data series containing the best values obtained in each run. The proposed HAOA was used as the control algorithm, and a Wilcoxon signed-rank test was executed on the above-mentioned data series. The obtained *p*-value in all cases was less than 0.05, (*p*-values were namely 0.03 vs. IHHO, 0.025 vs. HHO, 0.022 vs. SCA, 0.025 vs. AOA, 0.018 vs. PSO, 0.017 vs. DE, 0.026 vs. TLB, and finally, 0.028 vs. ChOA). It is possible therefore to conclude that the proposed HAOA method is statistically significantly better out of all the contending algorithms for both threshold values $alpha = 0.1$ and $alpha = 0.05$. The results of the Wilcoxon test are summarized in Table 9.

**Table 8.** Shapiro–Wilk test results.

| Methods | HAOA | AOA | SCA | IHHO | HHO | PSO | DE | TLB | ChOA |
|---------|------|-----|-----|------|-----|-----|-----|-----|------|
| *p*-value | 0.031 | 0.028 | 0.034 | 0.033 | 0.032 | 0.042 | 0.034 | 0.036 | 0.039 |

**Table 9.** Wilcoxon signed-rank test results.

| Methods | HAOA | AOA | SCA | IHHO | HHO | PSO | DE | TLB | ChOA |
|---------|------|-----|-----|------|-----|-----|-----|-----|------|
| *p*-value | N/A | 0.025 | 0.022 | 0.03 | 0.025 | 0.018 | 0.017 | 0.026 | 0.028 |

## 6. Conclusions

Fast diagnostics is crucial in modern medicine. The ongoing COVID-19 epidemic has shown how important it is to quickly determine whether or not a patient has been infected, and fast treatment is often the key factor to saving lives. This paper introduces a novel early diagnostics method to detect the disease from lungs X-ray images. The proposed model utilizes a novel HAOA metaheuristics algorithm, which was created by hybridizing AOA and SCA algorithms with a goal to overcome the deficiencies of the basic variants. The solutions in the proposed hybrid algorithm start by performing an AOA search procedure, and if the solution does not improve over the iterations, it will switch to the SCA search mechanism (controlled by the additional *trial* parameter). If the solution still does not improve, ultimately, it will be replaced by a quasi-reflective opposite solution, as defined by the QRL procedure.

The HAOA algorithm was put to test on a set of hard CEC2017 benchmark functions and compared to the results of the basic AOA and SCA and another cutting-edge metaheuristics algorithm. It can be concluded that the HAOA undoubtedly achieves a higher level of performance than the other eleven tested algorithms. After proving the superior performance on the benchmark functions, the algorithm was employed in the machine learning framework, consisting of the simple CNN used for feature extraction and an XGBoost classifier, where HAOA was used to tune the XGBoost hyperparameters. The model was named CNN–XGBoost–HAOA, tested on a large COVID-19 X-ray images benchmark dataset, and compared to eight other metaheuristics algorithms used to evolve the XGBoost structure. The proposed CNN–XGBoost–HAOA obtained predominant accuracy of almost 99.4% on this dataset, leaving behind all other observed models.

The contribution of the proposed research can be defined on three levels. First— a simple light-weight network was generated, that is easy to train, operates fast and achieves decent performance on the COVID-19 dataset, where the XGBoost classifier was used instead of fully connected layers. Second—AOA metaheuristics was improved and used in the model. Finally, the whole model has been adapted to the COVID-19 dataset. The limitations of the proposed work are closely bound to these three levels of contributions. First, it was possible to execute more detailed experiments with the hyperparameters of the simple neural network to begin with, and it was also possible obtain another light structure that could have an even better level of performance; however, this was out of the scope of this work. Second, each metaheuristics algorithm can be modified in an infinite number of theoretically possible improvements (minor modifications and/or hybridization), leading to the conclusion that in theory, the level of improvements of the basic AOA could be even higher without increasing the complexity of the algorithm. It was also possible to include other XGBoost parameters to the tuning process, as there are many of them, but it was not possible to cover all this with just one study. Finally, experiments were executed with just one dataset, which has been balanced. The experiments with imbalanced datasets were not executed, because addressing imbalanced datasets was not goal of presented study.

Based on these encouraging results, the future work will be centered around gaining even more confidence in the suggested model by testing it further on the additional real-life COVID-19 X-ray datasets before considering the practical implementation as a part of the system that could be used in the hospitals to help in early COVID-19 diagnostics.

**Author Contributions:** Conceptualization, M.Z., N.B. and B.N.; methodology, N.B., G.K. and M.M.; software, N.B. and M.Z.; validation, B.N., M.M. and N.S.; formal analysis, N.S.; investigation, N.B., M.Z. and M.A.; resources, M.A., B.N., N.B. and N.S.; data curation, M.Z., M.A. and N.B.; writing—original draft preparation, B.N., G.K. and M.Z.; writing—review and editing, N.B., B.N.

and G.K.; visualization, N.B., G.K. and N.S.; supervision, N.B.; project administration, M.Z. and M.M.; funding acquisition, B.N. and G.K. All authors have read and agreed to the published version of the manuscript.

**Funding:** This research is supported by Ministry of Education and Science of Republic of Serbia, Grant No. III-44006.

**Institutional Review Board Statement:** Not applicable.

**Informed Consent Statement:** Not applicable.

**Data Availability Statement:** Partial source code along with csv containing extracted features from CNN is available via the following Github URL: https://github.com/nbacanin/Electronics2022 (accessed on 12 November 2022).

**Conflicts of Interest:** All authors declare no conflict of interest.

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
