# Peer review of "Hybrid CNN and XGBoost Model Tuned by Modified Arithmetic Optimization Algorithm for COVID-19 Early Diagnostics from X-ray Images"

_electronics, doi:10.3390/electronics11223798_

Round 1

Reviewer 1 Report

This paper is well written and the technical contributions are evident. The authors have presented the proposed methodology comprehensively. Extensive simulation studies are also performed to investigate the performance of proposed work. There are some comments provided to further enhance the work:

1. Lines 13 and 14 - Please be specific to describe the performance of algorithm such as accuracy level, specificity and etc. 

2. Line 31 - "asground" looks like a typo.

3. Section 2.1 - The organization of this subsection needs to be improved. There are many paragraph with only one sentence. Please revise to make it more compact. 

4. Figure 1 - Needs to be enlarged. The fronts are too small to read.

5. Section 2.2 - What do you mean "t-tj objective function"?

6. Section 2.3 - Why swarm intelligence algorithms are highlighted in this section? AOA is categorized as Physics based algorithm. It is more appropriate to cover Physics based algorithm and some of its notable algorithm in this subsection. 

7. Figure 4 - Needs to be enlarged. The fronts are too small to read.

8. Table 5 - What is the CNN compared in this table? Is it LeNet-5 that is used for feature extraction? Authors should clarify this in the main text.

9. Figures 9 and 10 are too small and need to be enlarged.

10. The results of Shapiro table for single-problem analysis and Wilcoxon signed-rank test should be presented in table form.

11. Please discuss about the limitations of current work. 

12. Authors are strongly suggested to share the source code in platform such as Github to benefit the researchers that might be interested with this research area.

Reviewer 2 Report

1.       The contribution of this study should be revised to clearly highlight the novelty of this study. The proposed search strategies and operators for improving the AOA are not clear.

2.       The AOA suffers from a wide variety of deficiencies. It is recommended to show how the improved AOA is boost these efficiencies and then applied the proposed algorithm to CNN. Experimental evaluation is recommended.

3.       The title is “Modified Arithmetic Optimization Algorithm”, but the proposed section is “Hybrid arithmetic optimization algorithm”. please clarify.

4.       The proposed method section is so weak and cannot reflect the novelty of this study.

5.       The importance of this study is not clear. Why is the AOA selected for tuning the parameters of CNN?

6.       It is suggested to boost and classify the related work section. In the first section, the improved and new metaheuristic algorithms can be stated such as the enhanced whale optimization algorithm for medical feature selection: A COVID-19 case study, Starling murmuration optimizer, QANA, Diversity-maintained multi-trial vector differential evolution algorithm for non-decomposition large-scale global optimization and in the second section, the existing metaheuristic algorithms that are proposed for tuning the CNN.

7.       The canonical AOA suffers from low population diversity. How is this deficiency improved?

8.       The novelty of this study is not clear.

9.       The quality of Figure 4 should be improved.

10.   Why the convergence curve of all competitors is not visualized in Figure 5? Please clarify and revised this figure.

11.   The last column of Table 5 is not clear. The presentation of this table should be revised.

12.   The ROC plot in Figure 10 should be improved.

13.   The overall effectiveness of the proposed algorithm should be computed. The authors are recommended to compute and compare the overall effectiveness of algorithms.

14.   The authors are suggested to provide the discussion section and highlighted the main reason for the findings of this study.

Round 2

Reviewer 1 Report

For me, it is enjoyable to read the revised manuscript and responses provided by authors to address my comments. I like how the authors explain the action taken, particularly for comments #6 and 10. I would like to recommend this manuscript to be accepted for publication. Well done to the team and I am really looking forward to see this paper to be published online. All the best to Mr Nikola Savanovic and hopefully you will complete your PhD studies soon.

Reviewer 2 Report

The revised manuscript is accepted.